# Proton-triggered rearrangement of the AMPA receptor N-terminal domains impacts receptor kinetics and synaptic localization

Josip Ivica[1,7], Nejc Kejzar[1,6,7], Hinze Ho[1,6,7], Imogen Stockwell[1], Viktor Kuchtiak[1,2], Alexander M. Scrutton[1], Terunaga Nakagawa[3,4,5] ✉ & Ingo H. Greger [1] ✉

AMPA glutamate receptors (AMPARs) are ion channel tetramers that mediate the majority of fast excitatory synaptic transmission. They are composed of four subunits (GluA1–GluA4); the GluA2 subunit dominates AMPAR function throughout the forebrain. Its extracellular N-terminal domain (NTD) determines receptor localization at the synapse, ensuring reliable synaptic transmission and plasticity. This synaptic anchoring function requires a compact NTD tier, stabilized by a GluA2-specific NTD interface. Here we show that low pH conditions, which accompany synaptic activity, rupture this interface. All-atom molecular dynamics simulations reveal that protonation of an interfacial histidine residue (H208) centrally contributes to NTD rearrangement. Moreover, in stark contrast to their canonical compact arrangement at neutral pH, GluA2 cryo-electron microscopy structures exhibit a wide spectrum of NTD conformations under acidic conditions. We show that the consequences of this pH-dependent conformational control are twofold: rupture of the NTD tier slows recovery from desensitized states and increases receptor mobility at mouse hippocampal synapses. Therefore, a proton-triggered NTD switch will shape both AMPAR location and kinetics, thereby impacting synaptic signal transmission.

Ionotropic glutamate receptors (iGluRs) mediate neurotransmission in response to presynaptic glutamate release at the majority of excitatory synapses in the brain[1]. AMPA glutamate receptors (AMPARs) enable the fast component of the postsynaptic response. They are ion channel tetramers consisting of the GluA1–GluA4 subunits in various combinations[2]. Receptors including the GluA2 subunit are Ca[2+] impermeable and are the most abundant throughout the forebrain. AMPAR organization at synapses is critical as both the receptor number[3,4] and their spatial arrangement determine the fidelity of synaptic

transmission and, therefore are substrates for synaptic plasticity[5–7]. Specifically, AMPAR proximity to presynaptic transmitter release sites has been suggested to be critical, as their low affinity for L-glutamate requires full exposure to the transmitter for optimal activation[8], while low glutamate concentrations (at locations distant from the glutamate transient) trigger receptor entry into nonconducting desensitized states. AMPARs laterally diffuse in the postsynaptic membrane[9], which provides a mechanism for both long-term and short-term synaptic plasticity (LTP and STP, respectively)[5]. The receptor location beneath

[1]Neurobiology Division, Medical Research Council (MRC) Laboratory of Molecular Biology, Cambridge, UK. [2]Institute of Physiology, Czech Academy of Sciences, Prague, Czech Republic. [3]Department of Molecular Physiology and Biophysics, Vanderbilt University School of Medicine, Nashville, TN, USA. [4]Center for Structural Biology, Vanderbilt University School of Medicine, Nashville, TN, USA. [5]Vanderbilt Brain Institute, Vanderbilt University School of Medicine, Nashville, TN, USA. [6]Present address: Department of Physiology, Development and Neuroscience, University of Cambridge, Cambridge, UK. [7]These authors contributed equally: Josip Ivica, Nejc Kejzar, Hinze Ho. ✉e-mail: terunaga.nakagawa@Vanderbilt.Edu; ig@mrc-lmb.cam.ac.uk

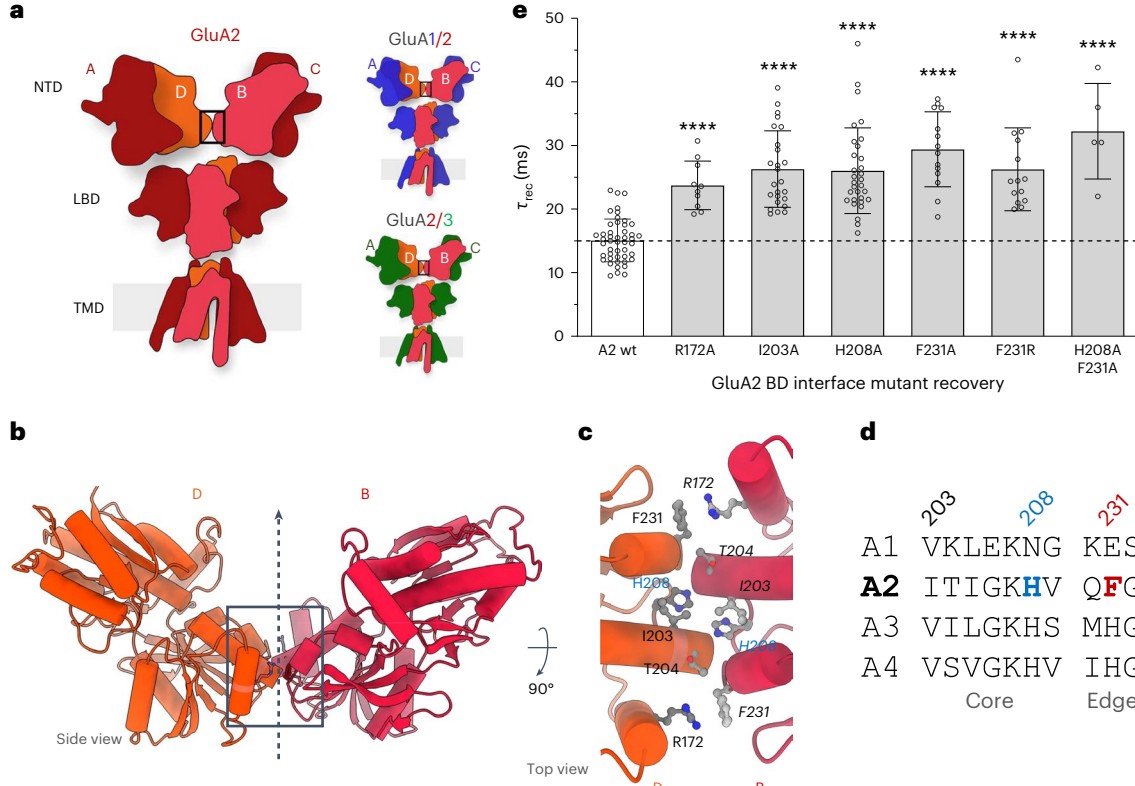

**Fig. 1 | A GluA2-specific NTD interface. a**, Left: schematic of an AMPAR tetramer, depicting the three domain layers and the inner BD subunits (light red), forming an interface between the GluA2 NTDs (boxed). Right: schematic of GluA2-containing AMPAR heteromers, where GluA2 localizes to the BD positions. **b**, Zoomed-in view of the GluA2 BD NTD interface (boxed); the vertical arrow denotes the two-fold symmetry axis. **c**, Top view of the further zoomed-in view of the GluA2 BD interface, showing the major interacting residues, including H208 and the cation−π interaction between R172 and F231. **d**, Sequence alignment of mouse AMPAR paralogs, showing divergence around the central histidine (blue) and the phenylalanine (brown) at the edges of the interface. **e**, Recovery from desensitization time constants for various mutants in GluA2 BD NTD interface, recorded at pH 7.4. Number of patches (GluA2 wt, $n = 47$; R172A, $n = 10$; I203A, $n = 25$; H208A, $n = 31$; F231A, $n = 14$; F231R, $n = 14$; H208A;F231A, $n = 5$). Bars show the mean and error bars denote the s.d. The effect of the substitution was determined by a one-way analysis of variance (ANOVA), $F_{(6,139)} = 26.36$, followed by Dunnett's multiple-comparisons test. ****$P < 0.0001$.

release sites is facilitated by cytoplasmic interactions with the postsynaptic density (PSD)[10–12] and extracellular anchoring within the synaptic cleft[13]. PSD associations have been studied extensively, while anchoring mechanisms through the AMPAR extracellular region (ECR) are currently less clear[14].

The AMPAR ECR is arranged as dimers of dimers and is two layered[15]: the membrane-proximal ligand-binding domain (LBD) coordinates the agonist L-glutamate, while the distal N-terminal domain (NTD) drives subunit assembly[16–18] and fulfills an incompletely understood anchoring role that supports both baseline transmission and synaptic plasticity[19–23]. The NTD sequence divergence between AMPAR subtypes enables association with different protein partners and highlights the intrinsic importance of the NTD tier in organizing this 'anchoring platform'[14,24]. In the predominating GluA2-containing AMPARs, the NTDs of two GluA2 subunits locate to the inner ('BD') positions of the receptor heterotetramer[25,26]. As a consequence, the receptor assembly is stabilized through a GluA2-specific NTD interface between the BD subunits (Fig. 1a). This interface is absent in GluA1, resulting in greater mobility of the NTD dimers, with consequences for gating and synaptic signaling[27]. Moreover, a mutation that destabilizes the GluA2 NTD interface (F231A) slows gating kinetics, reduces postsynaptic currents and impacts STP[27], thereby underscoring the role of a compact, tetrameric NTD tier for AMPAR function.

The synaptic cleft environment is subject to activity-dependent pH fluctuations[28]. Proton concentrations change rapidly in response to intense synaptic activity, resulting from the release of acidic synaptic vesicles (pH 5.3–5.7)[28–31], and substantial acidification occurs

in pathological states including ischemia and stroke[28]. Indeed, some proton-gated acid-sensing channels (ASICs) are activated at pH values below pH 5.0, such as ASIC2a (ref. 32), which resides in dendritic spines[33]. Protons impact the structure and function of a variety of other synaptic components, such as voltage-gated $Ca^{2+}$ channels, GABA-A receptors and iGluRs. NMDA-type iGluRs (NMDARs) are tonically inhibited at physiological pH (pH 7.0–7.4), while alkalinization of the cleft boosts the NMDAR response[34]. High proton concentrations also reduce AMPAR activity, lowering their open probability and accelerating desensitization rates through undefined mechanisms[35,36]. While the modulatory action of protons on various receptors is well established, the actual sites of proton sensing are either not known or are highly distributed, as in NMDARs[1].

Using a combination of all-atom molecular dynamics (MD) simulations, patch-clamp electrophysiology and cryo-electron microscopy (cryo-EM), we now show that protons reorganize the GluA2 ECR, rupturing the NTD tier akin to the F231A substitutions[27]. MD simulations implicate H208, buried in the center of the NTD BD interdimer interface, as a key proton sensor: H208 protonation breaks interfacial hydrogen bonds (H-bonds), which causes a rotation of the histidine side chain away from the interface, thereby destabilizing the tetrameric NTD tier. Protons also slow GluA2 recovery from desensitized states, as do various substitutions within the NTD BD interface. Therefore, acidification of the synapse triggers desensitized conformations that may facilitate detachment from synaptic anchor points and increase receptor diffusion, with consequences for synaptic signaling. This scenario is supported by fluorescence recovery after photobleaching

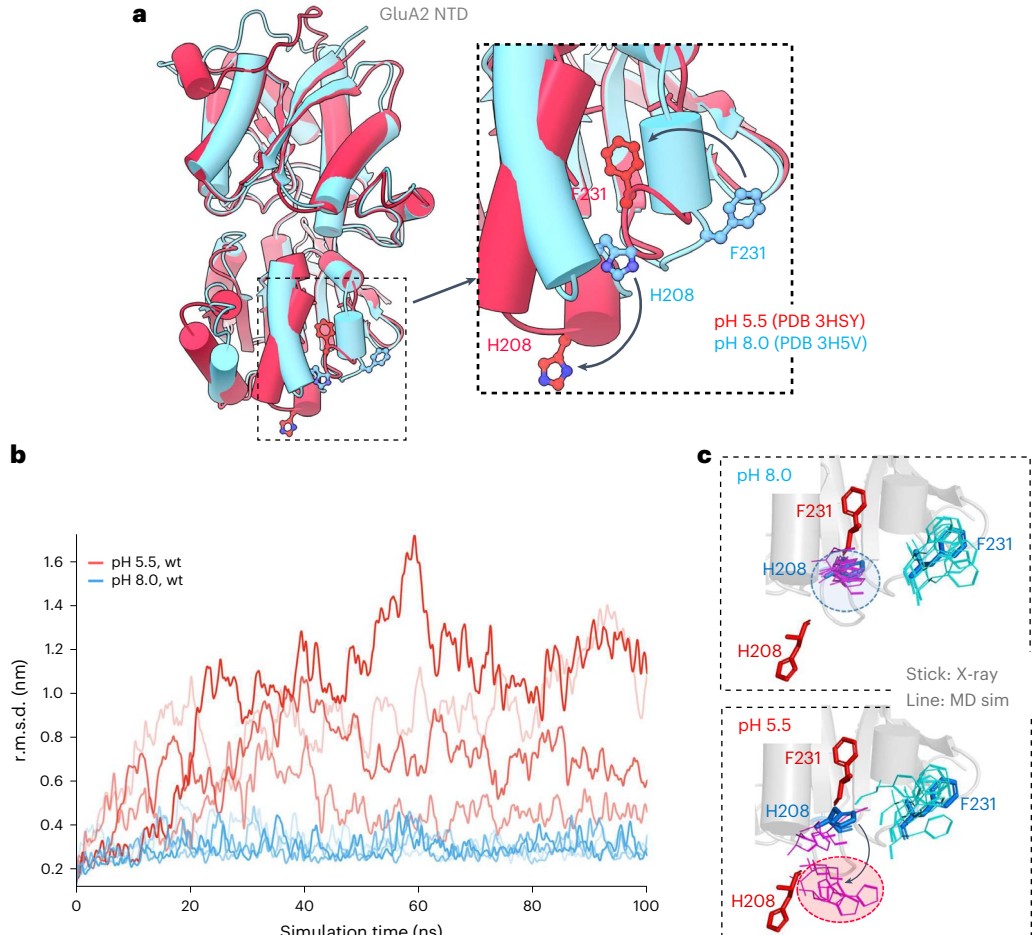

**Fig. 2 | pH sensitivity of the GluA2 BD NTD interface. a**, Overlay of two (monomeric) GluA2 NTD crystal structure models, determined at acidic (red) and neutral (blue) conditions. The pH-dependent rearrangements of H208 and F231, away from the interface (bend, black arrows), are boxed and are magnified in the right panel. **b**, MD simulation of the NTD tetramer (PDB 3H5V), showing large r.m.s.d. changes at acidic pH (red traces) but not at neutral pH (blue).

Four runs are shown for each. **c**, Positions of the H208 and F231 side chains (line format) at the end of the simulations, mapped onto the crystal structures (stick format). Red sticks, PDB 3HSY; blue sticks, PDB 3H5V. Simulation conditions at pH 8.0 (top) and pH 5.5 (bottom) are shown. H208 is mostly flipped downward throughout the runs at acidic pH (red ellipsoid) but not at neutral pH (blue ellipsoid). F231 largely remains unchanged (matching PDB 3H5V).

(FRAP) experiments demonstrating that NTD splaying accelerates GluA2 receptor mobility at the postsynapse.

## Results

### GluA2 stabilizes the NTD tier through a BD chain interface

AMPAR tetramers harbor two conformationally distinct subunit pairs designated AC and BD[15], which, akin to NMDARs[1], contribute differently to gating. The GluA2 subunit, located in the BD positions in GluA2-containing AMPARs[25,26], governs gate opening and also stabilizes the receptor's Y-shaped architecture through an interface between the two NTD dimers (Fig. 1a,b, boxed, and Extended Data Fig. 1a). Despite its relatively small size (400 Å solvent-accessible surface area), this GluA2-specific NTD interface largely maintains a compact conformation of the receptor in structural snapshots of the gating cycle[37–40]. However, the BD interface is absent when the isolated GluA2 NTD is assessed by multiangle light scattering (MALS), even at high protein concentration (1–2 mg ml⁻¹), where only NTD dimers but no tetramers are apparent (Extended Data Fig. 1b), consistent with their high (low nanomolar) dimer affinity[18,41]. Hence, the NTD interdimer contacts apparent in the intact receptor must be maintained by the LBD and transmembrane domain (TMD) tiers; they are of low affinity and, thus, transient.

The two-fold symmetric BD NTD interface is stapled together by cation–π interactions between R172 and F231 on either end that are unique to GluA2 and is stabilized in its center by van der Waals contacts and H-bonds, including those between the H208 side chain and the I203 main chain (Fig. 1c,d and Extended Data Fig. 1c)[42]. The F231A substitution ruptures the interface and splays the NTD dimers apart, which leads to slowed recovery from nonconducting, desensitized states[27]. We document the relationship between recovery kinetics and BD interface stability using patch-clamp electrophysiology of GluA2 BD interface mutants expressed in HEK293 cells (R172A, I203A, H208A, F231R and H208A;F231A). When rapidly applying L-glutamate to outside-out membrane patches, all mutants collectively led to a ~2-fold slowing of desensitization recovery (Fig. 1e) but had overall little effect on desensitization entry kinetics (Extended Data Fig. 2a).

### The GluA2 BD NTD interface is pH sensitive

Through a comparison of existing GluA2 NTD crystal structures, which were determined under various pH conditions, we noticed that the BD interface was intact in alkaline crystallization buffers (for example, Protein Data Bank (PDB) 3H5V and PDB 3N6V)[18,42]. and closely mirrored its organization in intact receptor structures. By contrast, at acidic pH conditions (for example, PDB 3HSY and PDB 3H5W)[18,42,43],

**Table 1 | Kinetic parameters of GluAx constructs**

| | Peak (pA) | | $\tau_{w, desensitization}$ (ms) | | $\tau_{recovery}$ (ms) | | Steady state | |
|---|---|---|---|---|---|---|---|---|
| | pH 7.4 | pH 5.5 | pH 7.4 | pH 5.5 | pH 7.4 | pH 5.5 | pH 7.4 | pH 5.5 |
| A2 wt | 468±371 (27) | 299±244 (27) | 14.3±2.3 (27) | 3.7±1.2 (27) | 15.6±2.9 (22) | 28.8±3.6 (22) | 8.6±2.8 (27) | 1.6±1.1 (27) |
| A2$_{delNTD}$ (n=7) | 546±407 | 378±320 | 18.9±1.2 | 4.9±0.3 | 17.7±2.2 | 21.3±1.5 | 11.9±2.3 | 3.7±2.0 |
| A2$_{NTD\_A1}$ (n=8) | 846±303 | 521±228 | 13.3±1.5 | 4.1±0.7 | 27.0±2.5 | 33.9±3.3 | 8.6±1.5 | 3.3±1.1 |
| A2$_{H208M}$ (n=6) | 331±49 | 200±39 | 10.6±2.0 | 2.9±0.7 | 29.1±1.9 | 37.0±3.1 | 6.1±2.2 | 2.4±1.4 |
| A2$_{F231A}$ (n=9) | 254±108 | 155±73 | 11.1±1.3 | 3.0±0.4 | 27.2±2.9 | 36.3±4.6 | 7.8±2.0 | 3.0±1.5 |
| A1wt (n=7) | 329±159 | 228±129 | 5.1±0.8 | 3.0±0.7 | 138±14 | 202±36 | 3.8±1.4 | 1.9±0.9 |
| A1$_{NTD\_A2}$ (n=7) | 315±126 | 171±53 | 4.6±1.0 | 2.1±0.2 | 116±16 | 284±46 | 1.3±0.8 | 0.4±0.3 |

Values are presented as the mean±s.d.

a drastic reorientation of two interface residues, H208 and F231, is apparent; the H208 side chain swings down toward the LBD, while F231 is buried within a hydrophobic pocket, inside the NTD core (Fig. 2a and Extended Data Fig. 1d). As these two side chains point away from the interface-forming region at acidic pH, they would contribute to BD interface destabilization.

To assess whether pH influences interface stability, we subjected the NTD dimer arrangement from PDB 3H5V to all-atom MD simulations at both pH 5.5 and pH 8.0. We conducted four independent 100-ns simulations for each condition and determined the root-mean-square deviation (r.m.s.d.) throughout the simulations; while the runs closely reflected the starting structure under alkaline conditions, they exhibited drastic deviations at acidic pH (Fig. 2b). When comparing the structures at the start and end of the simulations, we noted a global rearrangement of the NTD dimers at pH 5.5 but not at pH 8.0. Moreover, the H208 side chain was flipped into a downward conformation at pH 5.5, closely mimicking the crystal structures determined at acidic pH (Fig. 2c), whereas, at pH 8.0, the H208 side chain remained in its interfacing conformation (Extended Data Fig. 1e and Supplementary Videos 1 and 2). We observed no notable changes of the F231 side chain in the acidic runs, indicating that the F231 conformational change happens over longer time scales. Hence, protonation of H208 at pH 5.5 is expected to induce charge repulsion, break the H-bonds with I203 (Extended Data Fig. 1c) and destabilize the NTD tetramer. As NTD dimer splaying is consequential for synaptic AMPARs[27], we carried out a series of functional and structural studies to better understand the underlying mechanism.

**Low pH impacts AMPAR gating**

We performed patch-clamp recordings of GluA2 expressed in HEK293 cells and monitored current responses at different pH conditions. We observed a reduction in the peak amplitude and a substantial acceleration of desensitization kinetics when switching from pH 7.4 to 5.5, with pH 6.4 producing an intermediate effect (Extended Data Fig. 2b–d and Table 1). In addition, a ~2-fold slowing of recovery from the desensitized state was apparent, similar in magnitude to the substitutions in the NTD BD interface (Extended Data Fig. 2b,e and Fig. 1e)[27]. Together with our MD data (Fig. 2b,c and Supplementary Videos 1 and 2) and the results below, we considered that slowed recovery reflected an instability of the NTD tier at low pH.

We repeated these recordings in the presence of the AMPAR auxiliary subunit TARP-γ2 (transmembrane AMPAR regulatory protein γ2). GluA2-containing receptors are structurally stabilized by their association with TARPs, while TARP-free receptors (which are unlikely to exist in the brain[44,45]) are more susceptible to NTD splaying, particularly under desensitizing conditions[46]. With GluA2–TARPγ2, the accelerated desensitization entry and slowing of recovery under acidic conditions closely resembled the TARP-free recordings (~2-fold slowing from 15.6 ± 2.9 ms at pH 7.4 to 28.8 ± 3.6 ms at pH 5.5) (Fig. 3a,b

and Table 1). Similarly, we observed a reduction in both the peak and the equilibrium response (Table 1 and Extended Data Fig. 3a), as well as accelerated entry into deactivation (Extended Data Fig. 3b,c) and a rightward shift in the L-glutamate dose–response relationship (from half-maximal effective concentration (EC$_{50}$) = 0.27 ± 0.04 mM at pH 7.4 to 1.84 ± 0.60 mM at pH 5.5) (Extended Data Fig. 3d), together suggesting that protons modulate a range of parameters of native-like AMPAR–TARP complexes. Further characterization by nonstationary noise analysis revealed that the reduction in peak currents at acidic pH was because of a reduced channel open probability, with no change in conductance (Extended Data Fig. 3e,f), consistent with earlier data[35,36].

To assess a potential role of the NTD in pH-dependent desensitization recovery, we next performed recordings of GluA2 NTD mutants. GluA1, which lacks the BD NTD interface[27], was assessed alongside. NTD-deleted GluA2 (GluA2$_{delNTD}$) exhibited strong proton-mediated acceleration into desensitization at pH 5.5, closely matching the GluA2 wild type (wt) (Fig. 3d and Table 1) but the slowing of recovery from desensitization was not as pronounced in GluA2$_{delNTD}$ (1.2-fold versus 1.85-fold for GluA2 wt) (Fig. 3c). Similarly, the recovery kinetics of the NTD-splayed GluA2$_{F231A}$ mutant was less sensitive to protons. Replacement of the GluA2 NTD with that of GluA1 also rendered the chimeric receptor (GluA2$_{NTDA1}$) less sensitive to pH, which more closely resembled GluA2$_{delNTD}$ and GluA2$_{F231A}$ than GluA2 wt (Fig. 3c and Table 1). By contrast, the reverse NTD swap, which transplanted the GluA2 NTD onto GluA1, substantially increased the proton sensitivity of the chimera (GluA1$_{NTDA2}$) (2.4-fold versus 1.5-fold for GluA1 wt), suggesting that GluA2 confers greater pH sensitivity onto GluA1 (Fig. 3c). Taken together, these data imply that an intact GluA2 NTD BD interface contributes to proton modulation of desensitization recovery.

**Role of GluA2 H208 in desensitization recovery**

To investigate the role of H208 in proton regulation, we recorded various substitutions at position 208. We found that any residue other than histidine slowed recovery, highlighting an optimally evolved BD NTD interface on the one hand and its transient nature on the other (Fig. 3d). For example, substitution to phenylalanine (H208F), where the phenylalanine side chain is expected to 'fill' the space occupied by histidine and engage the opposite NTD dimer through van der Waals contacts, resulted in reduced recovery rates (Table 1). Similarly, T208 has a capacity to form an H-bond across the interface but the H208T substitution similarly attenuated recovery rates. The integrity of the BD NTD interface was recapitulated with AlphaFold2 (ref. 47), which predicted interface contacts in both GluA2 homomer and GluA1/2 heteromeric NTDs but not in GluA1 homomers (Extended Data Fig. 4a–c), thus matching experimental data[27]. The H208 substitutions similarly destabilized the GluA2 interface, as did the F231A substitution (Extended Data Fig. 4a–c), and these findings could be extended

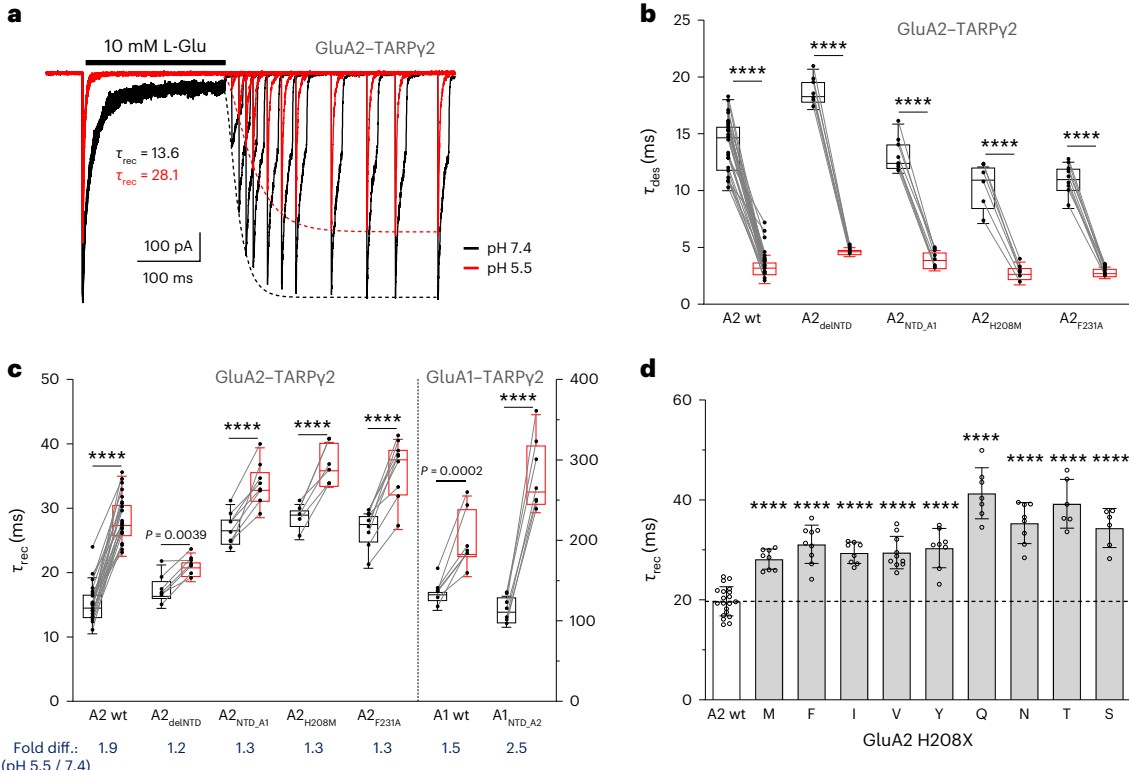

**Fig. 3 | Proton sensitivity of AMPAR gating kinetics. a**, Example trace of paired-pulse current from a membrane patch containing GluA2–TARPγ2 at pH 7.4 (black) and pH 5.5 (red) at −60 mV. The dashed lines represent Hodgkin–Huxley fits to recovery data ($\tau$ = 13.6 ms at pH 7.4; 28.1 ms at pH 5.5). **b**, Box plots showing the entry into desensitization of five GluA2–TARPγ2 constructs. pH 7.4, black boxes; pH 5.5, red boxes. Paired values were obtained from the same patch (lines). Number of patches: wt, $n$ = 27; A2$_{delNTD}$, $n$ = 7; A2$_{NTD\_A1}$, $n$ = 8; A2$_{H208M}$, $n$ = 6; A2$_{F231A}$, $n$ = 9. Boxes show the 25th and 75th percentiles, whiskers enclose points within 1.5× the interquartile range from the 25th and 75th percentiles and the horizontal line denotes the median. Data points are plotted as closed circles where the smallest and the largest data points are the minimum and maximum. A two-way ANOVA revealed the effects of pH ($F_{(1,52)}$ = 1,049, ****$P$ < 0.0001), the NTD ($F_{(4,52)}$ = 22.24, ****$P$ < 0.0001) and their interaction ($F_{(4,52)}$ = 12.16, ****$P$ < 0.0001). **c**, Box plots of the recovery from desensitization for five GluA2 constructs with modifications in the NTD and two GluA1 constructs (GluA1 wt and GluA1$_{NTD\_A2}$; all expressed with TARPγ2). Number of patches: A2 wt, $n$ = 27; A2$_{delNTD}$,

$n$ = 7; A2$_{NTD\_A1}$, $n$ = 8; A2$_{H208M}$, $n$ = 6; A2$_{F231A}$, $n$ = 9; A1 wt, $n$ = 7; A1$_{NTDA2}$, $n$ = 7. Boxes and whiskers are as in **b**. Data obtained from GluA2 and GluA1 constructs are separated by the dashed line and statistical tests were conducted separately. For GluA2, a two-way ANOVA revealed the effects of pH ($F_{(1,47)}$ = 279.9, ****$P$ < 0.0001), the NTD ($F_{(4,47)}$ = 47.48, ****$P$ < 0.0001) and their interaction ($F_{(4,47)}$ = 14.94, $P$ < 0.0001). For GluA1, a two-way ANOVA revealed the effect of pH ($F_{(1,12)}$ = 184.4, ****$P$ < 0.0001), no effect of substitution ($F_{(1,12)}$ = 4.543, $P$ = 0.0544) and an effect of their interaction ($F_{(1,12)}$ = 37.29, ****$P$ < 0.0001). **d**, The desensitization recovery values of GluA2 H208X mutants recorded in whole-cell configuration without the auxiliary subunit at pH 7.4. Number of cells: A2 wt, $n$ = 20; H208M, $n$ = 8; H208F, $n$ = 9; H208I, $n$ = 8; H208V, $n$ = 10; H208Y, $n$ = 8; H208T, $n$ = 8; H208Q, $n$ = 7; H208N, $n$ = 8; H208T, $n$ = 6; H208S, $n$ = 6. Bars show the mean and error bars denote the s.d. The effect of the substitutions was determined by a one-way ANOVA, $F_{(9,80)}$ = 34.39, followed by Dunnett's multiple-comparisons test. ****$P$ < 0.0001.

when modeling interface stability and binding affinity against a spectrum of GluA2 H208 substitutions using DynaMut2 (Extended Data Fig. 4d–f)[48].

We next subjected GluA2$_{H208M}$ to further functional analysis. This mutant exhibited a smaller effect on recovery kinetics at neutral pH compared to most other variants at position 208 (Fig. 3d). When switching to acidic pH in excised patch recordings, GluA2$_{H208M}$ slowed recovery relative to pH 7.4 but to a lesser extent than the wt. Hence, GluA2 proton modulation is reduced in the absence of H208. In fact, GluA2 closely resembled GluA2$_{F231A}$ (Fig. 3c) and GluA2$_{delNTD}$ (lacking the H208 'proton sensor'), suggesting that protonation of the LBDs contributes to the slowing of recovery in GluA2$_{H208M}$. To gain deeper insight into the importance of His208 in proton sensing, we conducted refined all-atom MD simulations where we controlled H208 protonation.

## MD simulations targeting GluA2 H208

We selectively protonated the H208 side chains and simulated the NTD tetramer at otherwise alkaline pH (Fig. 4a,b); vice versa, we deprotonated H208 and ran the simulation under acidic pH conditions (Fig. 4c,d). Directly protonating both histidines resulted in a dramatic

destabilization of the NTD tetramer, which is apparent when compared to overall alkaline conditions (Figs. 2b and 4a, gray insert). We even observed a complete dissociation of the NTD dimers in one of the runs (Fig. 4a, asterisk). These results underline the strategic role of H208 in proton sensing. In the reverse setting, deprotonation of the H208 side chains in an acidic pH background resulted in a subtle stabilization of the structure when compared to overall acidification (Fig. 4c), indicating that H208 is a central but not the sole determinant of interface destabilization at acidic pH, as discussed below. Inspecting the structures after the run revealed flipped H208 side chains when directly protonated (Fig. 4b), whereas the interfacing conformation was retained when H208 was deprotonated (Fig. 4d). Hence, solely protonating the two interfacing H208 residues caused a major destabilization of the otherwise intact NTD BD interface.

We extended two simulations (at pH 5.5 and 8.0, respectively) to 300 ns to divulge potential longer-term conformational changes. While the system behaved similarly to the shorter (100 ns) runs at pH 8.0, more dramatic rearrangements of the NTD dimers were evident at pH 5.5, which were again accompanied by a flipped-down H208 side chain (Extended Data Fig. 5a,b).

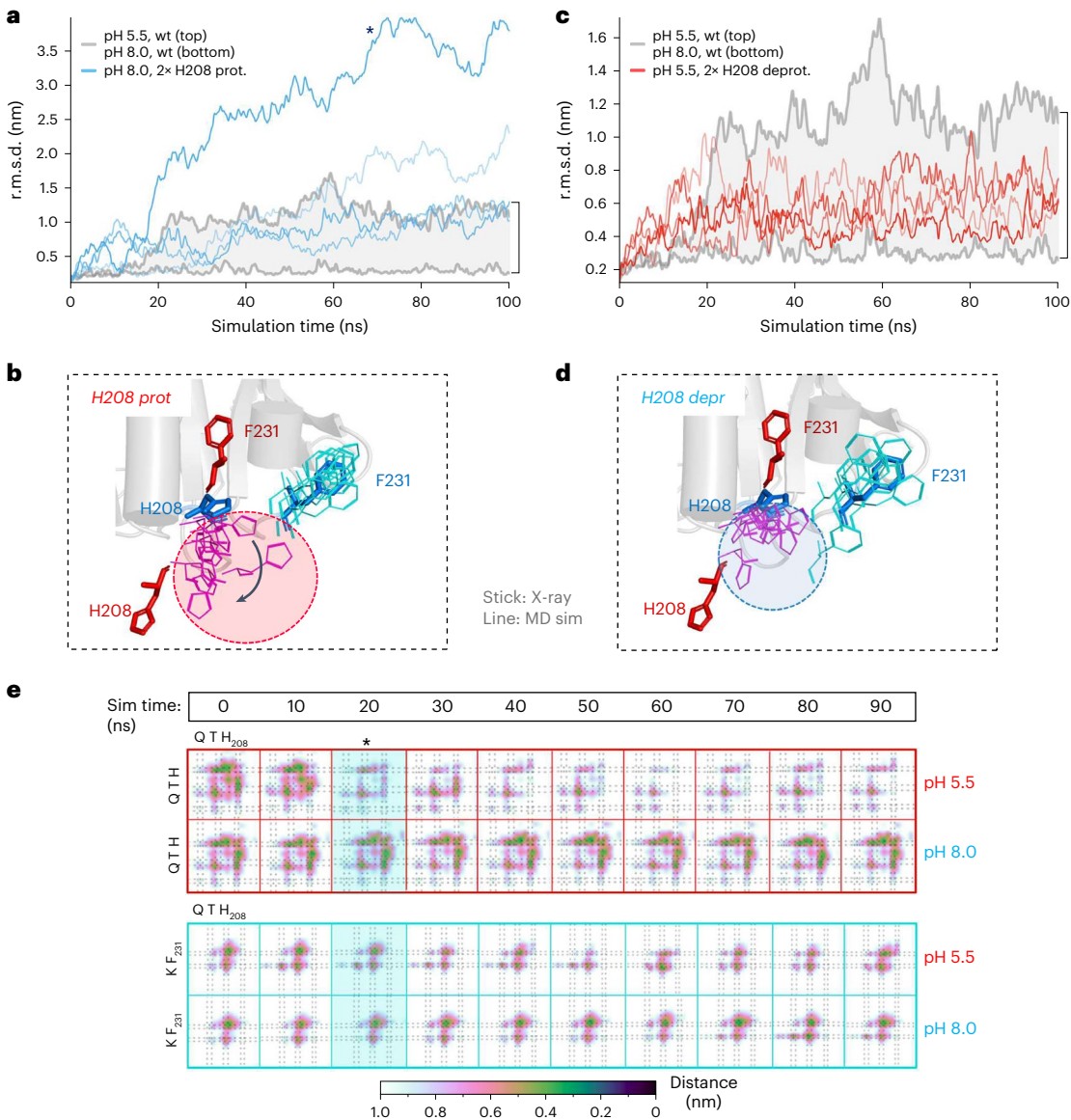

**Fig. 4 | H208 is a proton sensor. a**, Four MD simulations showing r.m.s.d. changes of the NTD tetramer (PDB 3H5V) run at neutral pH but with both H208 side chains protonated (blue traces). The gray zone shows the entire r.m.s.d. span from the runs shown in Fig. 2b; the two runs outside this zone (asterisk) exhibited rupture of the NTD tetramer. **b**, Directly protonated H208 side chains (purple lines, circled) in otherwise neutral pH conditions swing downward toward the position of the acidic crystal structure (PDB 3HSY, red sticks). The F231 side chains (cyan lines) remain unchanged. **c**, MD runs where both H208 side chains were deprotonated and run under acidic conditions, showing stabilization of the NTD tetramer compared to complete acidic conditions (upper gray boundary). **d**, As in **b**, showing that deprotonating H208 (purple lines, circled) stabilizes H208 in an interfacing position in an otherwise acidic environment. **e**, Temporal evolution of contact distances (horizontal bar) across the BD interface. QTH labels refer to Q201, T204 and H208. Contacts ruptured at 20 ns at acidic pH (5.5) but not at pH 8.0 (red rows). In the cyan rows, weaker changes can be seen between T204 (Q**T**H) and F231 (K**F**), which occurred at ~50 ns and were absent at pH 8.0.

Lastly, we generated interaction plots of interfacing residues and computed the s.d. of pairwise residue distances across the BD interface[38,49]. The largest fluctuations at acidic pH occurred between the interfacing H208 residues, whereas such changes were absent in alkaline conditions. Moreover, switching protonation states (that is, protonating H208 in alkaline pH and vice versa) reversed this trend (Extended Data Fig. 5c). To complement these data, we also computed average residue encounter times, where notable encounters of the opposed H208 side chains were evident in the intact interface at alkaline pH but were mostly absent at pH 5.5. This pattern was again largely reversed when switching H208 protonation states (Extended Data Fig. 5d). We also

observed changes between the T204 and F231 regions of the interface (Fig. 1c and Extended Data Fig. 5c,d) but these were less pronounced than the H208–H208 interactions and implied more global rearrangements.

When tracking the behavior of interface residues over simulation time, we detected a large change between the H208 environments starting at ~20 ns (Figs. 2b and 4a,e), which was accompanied by a flipped-down H208 side chain (Fig. 2c). Changes in the interaction between H208 and F231 were less pronounced and followed afterward (Fig. 4e). This suggests that the contacts between the interfacing histidines are not only the source of the largest destabilization but also appear to be its trigger.

**Table 2 | Cryo-EM data collection, refinement and validation statistics**

| GluA2*flip*(Q)–TARPγ2 | pH 8 consensus LBD–TMD–STG; EMD-44232, PDB 9B5Z | pH 8 consensus TMD–STG; EMD-44233, PDB 9B60 | pH 8 class 1 LBD–TMD–STG; EMD-44248, PDB 9B67 | pH 8 class 1 NTD; EMD-44249, PDB 9B68 |
|---|---|---|---|---|
| **Data collection and processing** | | | | |
| Microscope | Titan Krios G4 | Titan Krios G4 | Titan Krios G4 | Titan Krios G4 |
| Detector | BioQuantumK3 | BioQuantumK3 | BioQuantumK3 | BioQuantumK3 |
| Magnification | ×105,000 | ×105,000 | ×105,000 | ×105,000 |
| Voltage (kV) | 300 | 300 | 300 | 300 |
| Electron exposure (e⁻ per Å²) | 52.8 | 52.8 | 52.8 | 52.8 |
| Defocus range (μm) | −1.0 to −2.4 | −1.0 to −2.4 | −1.0 to −2.4 | −1.0 to −2.4 |
| Pixel size (Å) | 0.820 | 0.820 | 0.820 | 0.820 |
| Symmetry imposed | *C1* | *C1* | *C1* | *C1* |
| No. of micrographs | 21,898 | 21,898 | 21,898 | 21,898 |
| Final particle images (no.) | 1,108,462 | 1,108,462 | 47,782 | 47,782 |
| Map resolution (Å) | 2.71 | 2.57 | 3.39 | 3.60 |
| FSC threshold | 0.143 | 0.143 | 0.143 | 0.143 |
| Map resolution range (Å) | 2.49–4.33 | 2.43–4.40 | 3.03–6.33 | 3.49–6.61 |
| **Refinement** | | | | |
| Initial model used (PDB code) | 8FPG, 8FPK | 8FPG, 8FPK | 8FPG, 8FPK | 6U6I |
| Model resolution (Å) | 2.2/2.8 | 2.3/2.7 | 2.7/3.4 | 3.3/3.6 |
| FSC threshold | 0.5/0.143 | 0.5/0.143 | 0.5/0.143 | 0.5/0.143 |
| Map sharpening *B* factor (Å²) | −77.8 (−30)ᵃ | −72.3 | −87 | −79.9 |
| Model composition | | | | |
| Nonhydrogen atoms | 18,684 | 10,461 | 18,439 | 11,364 |
| Protein residues | 2,406 | 1,348 | 2,372 | 1,496 |
| Ligands | | | | BMA: 2 |
| | | | | NAG: 9 |
| *B* factors (Å²) | | | | |
| Protein | 86.44 | 49.87 | 39.29 | 41.11 |
| Ligand | | | | 64.49 |
| R.m.s.d. | | | | |
| Bond lengths (Å) | 0.003 | 0.002 | 0.003 | 0.003 |
| Bond angles (°) | 0.559 | 0.447 | 0.481 | 0.575 |
| Validation | | | | |
| MolProbity score | 1.80 | 1.55 | 1.42 | 1.55 |
| Clashscore | 7.97 | 4.52 | 6.28 | 6.35 |
| Poor rotamers (%) | 2.08 | 1.57 | 0.26 | 0.27 |
| Ramachandran plot | | | | |
| Favored (%) | 97.33 | 97.00 | 97.67 | 97.11 |
| Allowed (%) | 2.62 | 3.00 | 2.33 | 2.89 |
| Disallowed (%) | 0.04 | 0.00 | 0.00 | 0.00 |
| **GluA2*flip*(Q)–TARPγ2** | pH 8 class 12 LBD–TMD–STG; EMD-44251, PDB 9B6A | pH 8 class 12 NTD; EMD-44250, PDB 9B69 | pH 5.5 consensus LBD–TMD–STG; EMD-44234, PDB 9B61 | pH 5.5 consensus TMD–STG; EMD-44244, PDB 9B63 |
| **Data collection and processing** | | | | |
| Microscope | Titan Krios G4 | Titan Krios G4 | Titan Krios G4 | Titan Krios G4 |
| Detector | BioQuantumK3 | BioQuantumK3 | BioQuantumK3 | BioQuantumK3 |
| Magnification | ×105,000 | ×105,000 | ×105,000 | ×105,000 |
| Voltage (kV) | 300 | 300 | 300 | 300 |
| Electron exposure (e⁻ per Å²) | 52.8 | 52.8 | 55.6 | 55.6 |
| Defocus range (μm) | −1.0 to −2.4 | −1.0 to −2.4 | −1.0 to −2.4 | −1.0 to −2.4 |

**Table 2 (continued) | Cryo-EM data collection, refinement and validation statistics**

| GluA2*flip*(Q)–TARPγ2 | pH 8 class 12 LBD–TMD–STG; EMD-44251, PDB 9B6A | pH 8 class 12 NTD; EMD-44250, PDB 9B69 | pH 5.5 consensus LBD–TMD–STG; EMD-44234, PDB 9B61 | pH 5.5 consensus TMD–STG; EMD-44244, PDB 9B63 |
|---|---|---|---|---|
| Pixel size (Å) | 0.820 | 0.820 | 0.820 | 0.820 |
| Symmetry imposed | *C1* | *C1* | *C1* | *C1* |
| No. of micrographs | 21,898 | 21,898 | 19,684 | 19,684 |
| Final particle images (no.) | 48,399 | 48,399 | 813,615 | 813,615 |
| Map resolution (Å) | 3.35 | 3.69 | 2.81 | 2.76 |
| FSC threshold | 0.143 | 0.143 | 0.143 | 0.143 |
| Map resolution range (Å) | 3.04–6.55 | 3.42–6.92 | 2.66–6.06 | 2.65–4.63 |
| **Refinement** | | | | |
| Initial model used (PDB code) | 8FPG, 8FPK | 8FPG, 8FPK | 8FPG, 8FPK | 8FPG, 8FPK |
| Model resolution (Å) | 2.7/3.4 | 3.1/3.6 | 2.6/3.0 | 2.5/2.8 |
| FSC threshold | 0.5/0.143 | 0.5/0.143 | 0.5/0.143 | 0.5/0.143 |
| Map sharpening $B$ factor (Å$^2$) | −91.6 | −97.4 | −84.6 (−30)[a] | −83.7 |
| Model composition | | | | |
| Nonhydrogen atoms | 18,497 | 11,370 | 18,415 | 10,461 |
| Protein residues | 2,378 | 1,496 | 2,384 | 1,348 |
| Ligands | | BMA: 2 | | |
| | | NAG: 9 | | |
| $B$ factors (Å$^2$) | | | | |
| Protein | 32.63 | 62.33 | 119.67 | 48.25 |
| Ligand | | 91.91 | | |
| R.m.s.d. | | | | |
| Bond lengths (Å) | 0.003 | 0.002 | 0.005 | 0.002 |
| Bond angles (°) | 0.495 | 0.449 | 0.800 | 0.475 |
| Validation | | | | |
| MolProbity score | 1.43 | 1.47 | 2.05 | 1.73 |
| Clashscore | 7.42 | 5.52 | 10.98 | 4.77 |
| Poor rotamers (%) | 0.21 | 0.27 | 3.00 | 2.22 |
| Ramachandran plot | | | | |
| Favored (%) | 97.89 | 97.04 | 97.34 | 96.54 |
| Allowed (%) | 2.11 | 2.96 | 2.66 | 3.46 |
| Disallowed (%) | 0.00 | 0.00 | 0.00 | 0.00 |

| GluA2*flip*(Q)–TARPγ2 | pH 5.5 class 23 LBD–TMD–STG; EMD-44245, PDB 9B64 | pH 5.5 class 23 NTD; EMD-44245[e] |
|---|---|---|
| **Data collection and processing** | | |
| Microscope | Titan Krios G4 | Titan Krios G4 |
| Detector | BioQuantumK3 | BioQuantumK3 |
| Magnification | ×105,000 | ×105,000 |
| Voltage (kV) | 300 | 300 |
| Electron exposure (e$^-$ per Å$^2$) | 55.6 | 55.6 |
| Defocus range (µm) | −1.0 to −2.4 | −1.0 to −2.4 |
| Pixel size (Å) | 0.820 | 0.820 |
| Symmetry imposed | *C1* | *C1* |
| No. of micrographs | 19,684 | 19,684 |
| Final particle images (no.) | 29,945 | 29,945 |
| Map resolution (Å) | 3.56 | 5.90 |
| FSC threshold | 0.143 | 0.143 |
| Map resolution range (Å) | 3.27–7.57 | 5.90–9.04 |
| **Refinement** | | |
| Initial model used (PDB code) | 8FPG, 8FPK | 6U6I |

**Table 2 (continued) | Cryo-EM data collection, refinement and validation statistics**

| GluA2*flip*(Q)–TARPγ2 | pH 5.5 class 23 LBD–TMD–STG; EMD-44245, PDB 9B64 | pH 5.5 class 23 NTD; EMD-44245[c] |
|---|---|---|
| Model resolution (Å) | 3.1/3.6 | ND |
| FSC threshold | 0.5/0.143 | |
| Map sharpening *B* factor (Å$^2$) | −90.6 | −214 (−80)[b] |
| Model composition | | |
| Nonhydrogen atoms | 18,497 | ND |
| Protein residues | 2,378 | ND |
| Ligands | | |
| *B* factors (Å$^2$) | | |
| Protein | 97.32 | ND |
| Ligand | | |
| R.m.s.d. | | |
| Bond lengths (Å) | 0.003 | ND |
| Bond angles (°) | 0.639 | ND |
| Validation | | |
| MolProbity score | 1.63 | ND |
| Clashscore | 9.56 | ND |
| Poor rotamers (%) | 0.21 | ND |
| Ramachandran plot | | |
| Favored (%) | 97.38 | ND |
| Allowed (%) | 2.62 | ND |
| Disallowed (%) | 0.00 | ND |

Map sharpening *B* factors were determined by RELION postprocessing, unless otherwise noted. BMA, β-D-mannopyranose; NAG, 2-acetamido-2-deoxy-β-D-glucopyranose; ND, not determined because of low-resolution map. [a]A *B* factor of −30 was used to refine the atomic model (the *B* factor outside the parentheses was determined by RELION). [b]A *B* factor of −80 was used to interpret the map (a *B* factor of −214 was determined by RELION). [c]Deposited as an associated map.

## Cryo-EM structures of protonated GluA2–TARPγ2

We next determined cryo-EM structures of the GluA2–TARPγ2 receptor under two different pH conditions in an apo state (in the absence of L-glutamate) (Table 2 and Extended Data Fig. 6a–e). We coexpressed GluA2 and TARPγ2 (refs. 50,51) and purified the complex at pH 8.0 (Extended Data Fig. 6b). We chose pH 8.0 to ensure strict alkaline conditions for our structural studies and note that the GluA2–TARPγ2 kinetics at pH 8.0 matched that at pH 7.4, used in the functional studies above (Extended Data Fig. 6a). To protonate the receptor, we lowered the pH with citrate buffer immediately before vitrification (to pH ~5.5; Methods). As already apparent in the two-dimensional (2D) class averages, the signal encompassing the NTD tier was diffuse at low pH but was well defined throughout the receptor at pH 8.0 (Fig. 5a and Extended Data Fig. 6c). This difference was clearly reflected in the three-dimensional (3D) maps; the NTD tier was compact at pH 8.0 with the two dimers associated through the BD interface (comparable to previous structures[39,40,52]) (Extended Data Fig. 7a), while the BD interface was absent at pH 5.5 in most classes, resulting in highly heterogeneous NTD dimer conformations (for example, Fig. 5b–d and Extended Data Fig. 7b,c). To quantify this difference, we fitted atomic models into the cryo-EM envelopes (Methods) and measured the distance between the BD subunit NTDs. At pH 8.0, the BD chains were spaced apart by ~55 Å (center of mass (COM) distance) throughout the classes, while the BD distances ranged widely at pH 5.5; some NTD dimers adopted an upright conformation with a ruptured BD interface (class 31), while either one or both dimers bent toward the LBD tier in other classes (classes 19, 23 and 37), resulting in a range of COM distances between the BD chains (70–93 Å) (Fig. 5c,d and Extended Data Fig. 7d). The protonated receptor thus closely resembled the

structural heterogeneity of the GluA2 NTD point mutant F231A (Glu-A2$_{F231A}$), which also exhibited ~2-fold slowed recovery from desensitization (Fig. 3c)[27]. These results provide a structural correlate for the above data, illustrating that protons trigger instability and rupture of the NTD BD interface.

In GluA2$_{F231A}$–TARPγ2, NTD splaying was associated with rearrangements in the LBD tier[27]. Greater mobility of the LBDs was also observed in GluA1–TARPγ3, together suggesting that rupture of the NTD dimers is transmitted to the LBDs. When comparing the LBD sector between the two pH conditions, we noticed greater conformational heterogeneity of the LBDs at pH 5.5 versus pH 8.0, which is reflected in the local resolution maps of the consensus refinement (Extended Data Fig. 6f). Atomic models derived from the consensus refinements reveal differences in the LBD tier between the average LBD structures at pH 5.5 and pH 8.0, with larger displacements of the BD LBDs versus the AC LBDs in the pH 5.5 receptor (Extended Data Fig. 8a). These lead to an approximation of the LBD upper (D1) lobes between the BD subunits at pH 5.5 and to rearrangements of their G helices (Extended Data Fig. 8b, inset), which mark LBD conformations in different states of the gating cycle[53]. Other than the LBD tier, we also note subtle rotations of the TARPs; these are clockwise for the B′D′ TARPs but anticlockwise for the A′C′ TARPs when compared to the pH 8.0 structures and are somewhat reminiscent of those seen for auxiliary subunits between active-state and resting-state AMPARs[38,50,52]. Taken together, protonation induces multiple changes throughout the receptor assembly, including rupture of the NTD tier, rearrangements of the LBD pairs and rotational movements of the TARPs, culminating in reduced charge transfer through the receptor channel (Fig. 3 and Extended Data Fig. 3).

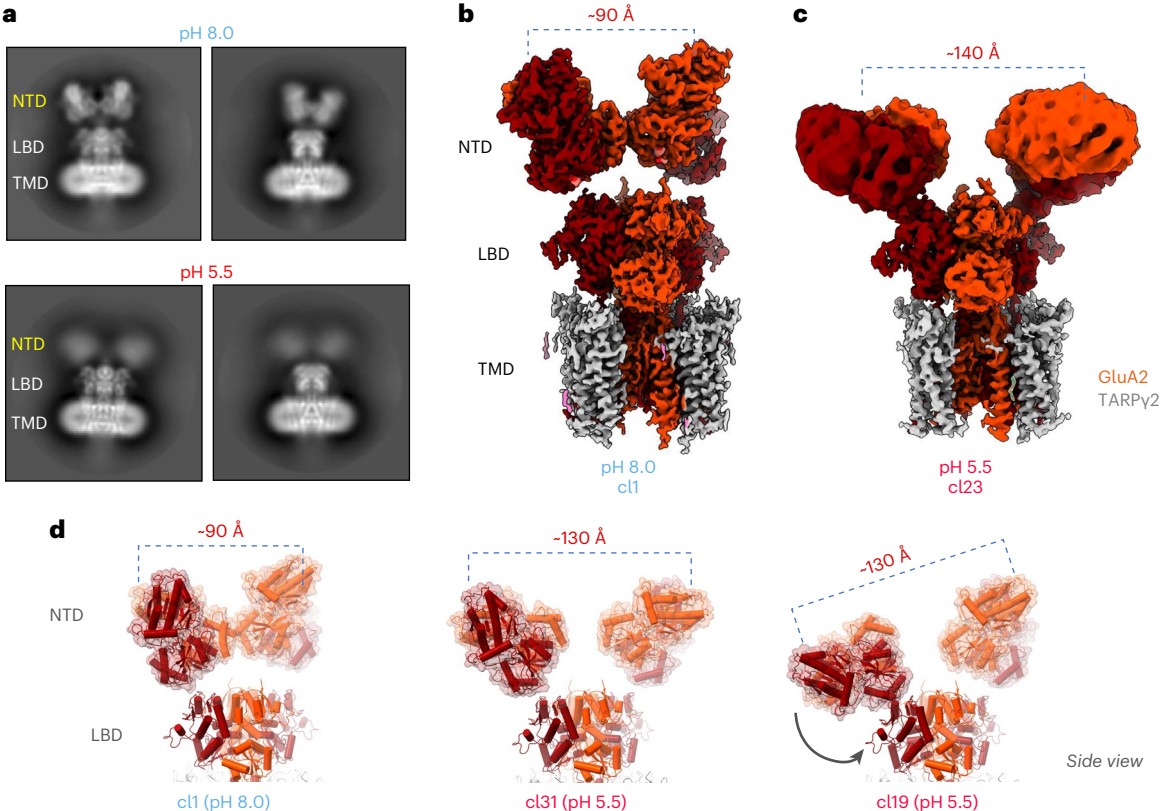

**Fig. 5 | GluA2–TARPγ2 cryo-EM structures at acidic and neutral pH. a**, The 2D class averages of the receptor collected at pH 8.0 (top) versus pH 5.5 (bottom). The NTD tier is well defined at neutral pH but not at acidic pH, where the signal for the NTD is blurred. Two views are shown for each. **b,c**, Cryo-EM maps of the pH 8.0 complex (**b**; class 1) and the pH 5.5 complex (**c**; class 23). The NTD dimers are splayed into a horizontal position at acidic pH, separating the N termini of the BD chains (orange) by ~140 Å (**c**) versus 90 Å for the compact BD NTDs determined at pH 8.0 (**b**). **d**, Atomic models of class (cl) 1 (pH 8.0) and of classes 31 and 19 (pH 5.5). NTD splaying at acidic pH separates the BD N termini as indicated above. Only the NTD and LBD tiers are shown.

## Increased mobility of NTD-splayed GluA2 at synapses

Lastly, we wanted to determine whether NTD splaying impacts AMPAR anchoring at the synapse and assayed receptor diffusion in primary hippocampal neurons using FRAP. We previously showed that the NTD enables synaptic retention of AMPARs as, unlike GluA2 wt, the GluA2$_{delNTD}$ mutant shows a near-complete recovery after photobleaching because of the replacement of freely diffusible (nonanchored) receptors[22]. To determine whether NTD conformation affects synaptic mobility, we first compared receptors with a compact (GluA2 wt) versus a splayed (GluA2$_{F231A}$) NTD tier, both fused at their N termini to supereccliptic pHluorin (SEP, a pH-sensitive variant of green fluorescent protein)[54]. We observed significant differences between GluA2 wt and GluA2$_{F231A}$, with the NTD-splayed mutant recovering more completely 10 min after bleaching (48% ± 0.03 for GluA2 wt–SEP; 61% ± 0.04 for GluA2$_{F231A}$–SEP) (Fig. 6a,b). Hence, GluA2$_{F231A}$ approached the behavior of GluA2 lacking its NTD (GluA2$_{delNTD}$), which exhibits substantially greater synaptic mobility than GluA2 wt[22]. Of note, this difference was not apparent when imaging receptors outside the synapse in the dendritic shaft, where receptors are known to diffuse freely (Fig. 6a). We then repeated the experiment at pH 5.5, which still permits the detection of SEP when using 405-nm excitation[54] and is expected to result in protonated (splayed) GluA2. At acidic pH, GluA2 wt indeed closely matched GluA2$_{F231A}$ imaged at pH 7.4 by recovering more completely to the levels before photobleaching (69% ± 0.02 for GluA2 wt–SEP at pH 5.5; 61% ± 0.04 for GluA2$_{F231A}$–SEP at pH 7.4) (Fig. 6a,b). This decrease in the immobile fraction of NTD-splayed receptors (either GluA2 wt at pH 5.5 or GluA2$_{F231A}$ at neutral pH) implies that rupture of the tetrameric NTD tier increases the lateral mobility of AMPARs at the synapse and,

together with a slowed desensitization recovery, contributes to shaping STP (Fig. 6c)[27].

## Discussion

We characterized the impact of protons on AMPAR structure and function. Low pH reduces receptor output (Fig. 3 and Extended Data Figs. 2 and 3) through previously unknown mechanisms[34–36]. Our cryo-EM structures reveal that protons lead to stark conformational changes throughout the GluA2–TARPγ2 complex that are most pronounced in the NTD tier. According to MD simulations and electrophysiology, protonation of H208 in the GluA2 BD NTD interface is a key trigger leading to interface destabilization and to splayed NTD dimers, which is associated with enhanced desensitization. While protons also target the LBD tier directly, as evident from strong, pH-dependent changes in GluA2$_{delNTD}$ desensitization (Fig. 3b), the lack of NTD BD contacts amplifies conformational rearrangements in the LBD and TMD sectors that are associated with desensitization. This scenario is supported by a previous structure–function analysis of the GluA2$_{F231A}$ point mutant, where loss of the BD NTD interface augments rearrangements of the LBDs, which is associated with slowed desensitization recovery[27]. Similarly, GluA1 lacks a BD interface (because of NTD sequence variation), resulting in substantial reorganization of both the NTD and the LBD tiers in response to agonist and entry into nonconducting states[27]. In both cases, GluA2$_{F231A}$ and GluA1, desensitization is accompanied by rupture of the LBD dimers into monomers, which is not seen in current desensitized GluA2 structures, where the LBDs rearrange but remain dimeric[38,55]. Together, this suggests that, by stabilizing the receptor assembly, the BD interface limits transitions into deeply desensitized states and enables their rapid recovery.

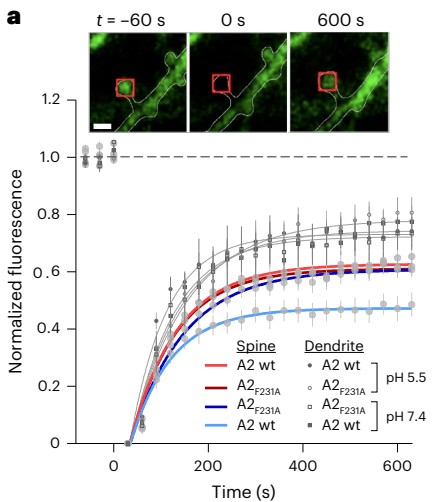

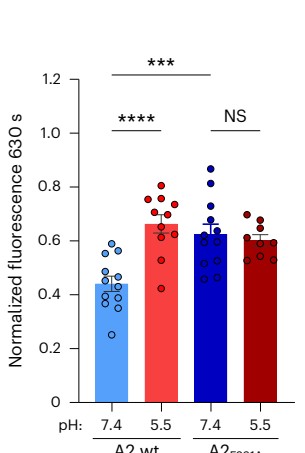

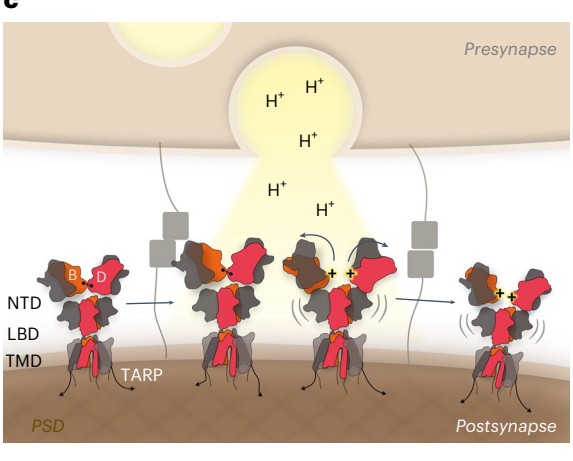

**Fig. 6 | NTD conformations determine GluA2 mobility at the synapse.**
**a**, Fluorescence recovery over time for GluA2 wt–SEP or GluA2$_{F231A}$–SEP expressed in dissociated hippocampal neurons, imaged at either pH 7.4 or pH 5.5. Normalized data were fit to a single exponential curve (time constant of fit: GluA2 pH 7.4, $\tau = 87$ s ($n = 12$ cells from three culture preparations); GluA2 pH 5.5, $\tau = 100$ s ($n = 11$); GluA2$_{F231A}$ pH 7.4, $\tau = 114$ s ($n = 12$); GluA2$_{F231A}$ pH 5.5, $\tau = 96$ s ($n = 9$)). Overlapping fluorescence recovery profiles for dendritic regions are shown in gray. The insets present example images with the dendrite outlined in white and the 2-μm$^2$ region of the spine imaged boxed in red. Scale bar,

2 μm. **b**. Fluorescence recovery 630 s after bleaching (GluA2 pH 7.4 = 0.44 ± 0.03 ($n = 12$ cells from three culture preparations), GluA2 pH 5.5 = 0.66 ± 0.03 ($n = 11$), GluA2$_{F231A}$ pH 7.4 = 0.63 ± 0.04 ($n = 12$) and GluA2$_{F231A}$ pH 5.5 = 0.60 ± 0.02 ($n = 9$). Statistical differences were determined by a one-way ANOVA. ****$P < 0.0001$ and ***$P = 0.0007$. **c**. Schematic depicting protonation of the GluA2 BD NTD (red) interface following presynaptic vesicle release, resulting in splaying of the GluA2 NTD and subsequent diffusion of desensitized receptors away from the release site (right) allowing renewal of the receptor population by resting or activatable receptors (left).

We further note that proton-induced splaying of the NTDs likely underlies the abundance of splayed NTDs observed in native AMPAR complexes in an earlier study, which was likely caused by the exposure to acidic uranyl formate stain[56].

Using FRAP imaging, we showed that GluA2 NTD splaying also increases receptor diffusion at the postsynapse. The GluA2 NTD has a synaptic anchoring role that supports baseline transmission and synaptic plasticity (both STP and LTP)[20,22,23]. We propose that this requires a compact, tetrameric NTD tier that optimally localizes the receptor for efficient signal transmission, likely proximal to glutamate release sites[7,14]. Loss of the BD interface in the GluA2$_{F231A}$ mutant results in reduced synaptic transmission and lowers paired-pulse facilitation (PPF)[27], a form of short-term synaptic enhancement, while complete removal of the GluA2 NTD mirrors these effects and further enhances lateral diffusion of GluA2$_{delNTD}$[22]. Together, this supports a synaptic anchoring role for the GluA2 NTD that requires an intact BD interface to locate the receptor at subsynaptic sites.

Diffusion trapping of AMPARs has been recognized as a postsynaptic mechanism for STP[57] and has a central role in the expression of LTP[58]. The STP mechanism posits that nonconducting (desensitized) receptors are replaced by resting (and, thus, readily activatable) receptors through lateral diffusion, enabling continued postsynaptic responses to closely spaced presynaptic inputs (that is, PPF)[57]. Blocking AMPAR lateral diffusion and, therefore, receptor replenishment at sites of glutamate release impairs both LTP and STP, the latter through reduced PPF[57]. How resting receptors are stabilized and desensitized receptors are released from their anchor has not been resolved. Engagement of synaptic cleft components, such as synaptic adhesion molecules[59] or secreted synaptic organizers, has been proposed on the one hand[13] and cytoplasmic interactions mediated by TARP auxiliary subunits with the PSD have been proposed on the other[10,11,60].

Here, we shine new light onto this regulation and propose the following model (Fig. 6c): synaptic vesicles release both L-glutamate and protons[28–31] onto AMPARs located within a trans-synaptic nanocolumn[6,7,12]. Glutamate binds to the LBDs to activate and desensitize the receptor, while protons target H208 in the NTD, causing rupture of the

BD interface. Through simultaneously targeting both ECR domains, LBD and NTD, vesicular release amplifies conformational changes and more effectively detaches receptors from their anchor(s) in the synaptic cleft[13,61–63], generating available slots for readily activatable receptors. These require a compact tetrameric NTD 'platform', enabled by GluA2 subunits in the BD position, for optimal anchoring. As vesicular acidification is expected to be very brief (hundreds of microseconds)[64] and is followed by prolonged alkalinization of the synaptic cleft[65], only AMPARs closely aligned with glutamate release sites would be subject to this proton-triggered mechanism. Because the GluA2 H208 proton sensor responded within tens of nanoseconds in our simulations (Fig. 4), we expect a sizeable number of AMPARs undergoing conformational transitions followed by their exit from the trans-synaptic nanocolumn.

GluA1 homomeric receptors, which exhibit a highly mobile NTD tier[27], are subject to different anchoring mechanisms that come into play during synaptic potentiation[66].

The synaptic diffusion and trapping mechanisms of other AMPAR subtypes, composed of the GluA3 and GluA4 subunits, are currently unclear. These receptors are enriched in the brain stem and in thalamic nuclei and are characterized by ultrarapid gating kinetics, facilitating high-frequency signal transmission[67]. Although both subunits encode a histidine at the position equivalent to GluA2 H208, they lack the GluA2 F231 equivalent and, thus, the cation–π interaction in the NTD BD interface critical for a stable, tetrameric NTD tier. Taken together, our data imply a unique regulation of the predominant, GluA2-containing AMPARs. These are endowed with a pH-dependent conformational switch—a compact NTD (at neutral pH) and a ruptured NTD (at acidic pH)—that responds to presynaptic activity and thereby tunes postsynaptic transmission and plasticity.

While this manuscript was in preparation, a paper presenting partly overlapping findings to ours appeared[68].

## Online content

Any methods, additional references, Nature Portfolio reporting summaries, source data, extended data, supplementary information,

# Article

acknowledgements, peer review information; details of author contributions and competing interests; and statements of data and code availability are available at https://doi.org/10.1038/s41594-024-01369-5.

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

## Methods

### Protein expression and purification

The complementary DNA (cDNA) constructs of a rat GluA2*flip*(Q) isoform, tagged with a FLAG epitope near its C terminus[39,51], and rat TARPγ2 (stargazin) were cloned into the DualTetON plasmid as described previously[69,50] to generate a plasmid named DualTetON-A2iQFLAG, which doxycycline-dependently expresses both proteins simultaneously. The two proteins were coexpressed without using any tether. A stable TetON HEK cell line was generated by cotransfecting DualTetON-A2iQFLAG and a plasmid that confers hygromycin resistance, using established methods[39,51,69]. A clone was isolated in the presence of 30 µM NBQX and 120 µg ml⁻¹ hygromycin. Clone3-#39 was chosen on the basis of its growth rate and the expression level of the complex and adapted to FreeStyle293 medium (Gibco, Thermo Fisher) in suspension.

Next, 1.2 L of a near-saturated suspension culture of clone3-#39 in FreeStyle293 medium supplemented with 30 µM NBQX and 1:500 diluted anticlumping agent (Gibco, Thermo Fisher, cat. no. 0010057DG) was used as a starting material. Cells were induced with 7.5 µg ml⁻¹ doxycycline, 1 mM sodium butylate and 1% fetal calf serum (FCS) for 28 h as described[39]. The subsequent procedures were conducted on ice or at 4 °C. Cells were centrifuged at 931$g$ for 10 min, washed with Dulbecco's PBS once and centrifuged again; the pellet was flash-frozen in liquid nitrogen for storage at −80 °C. Approximately 10–12 ml of frozen pellets were resuspended in Resuspend buffer (25 mM Tris-HCl pH 8.0, 150 mM NaCl, 2 mM TCEP, 15 µM NBQX and protease inhibitors: 1 mM PMSF, 10 µg ml⁻¹ aprotinin, 0.5 mM benzamidine, 1 µg ml⁻¹ pepstatin A and 5 µg ml⁻¹ leupeptin), making the final volume 90 ml. Then, 10 ml of 10× digitonin (25 mM Tris-HCl pH 8.0, 150 mM NaCl and 7.5% digitonin) was added and the mixture was nutated at 4 °C for 2.5 h to dissolve the membrane. The large debris was removed by low-speed centrifugation (3,000 r.p.m. for 10 min at 4 °C) and its supernatant was ultracentrifuged at 235,400$g$ in a 45Ti rotor (Beckman) for 1 h. The resulting supernatant was incubated in a batch with 1 ml of FLAG M2 agarose beads (Sigma) for 2 h. The beads were collected by centrifugation at 58$g$ for 5 min and transferred into an empty column. The beads were washed with four column volumes of wash buffer (0.03% glyco-diosgenin (GDN), 20 mM Tris-HCl pH 8.0 and 150 mM NaCl). The proteins were eluted using 6 ml of wash buffer containing 0.5 mg ml⁻¹ FLAG peptide. The eluate was concentrated down to 0.55 ml using Ultrafree 100-kDa molecular weight cutoff (MWCO) ultrafiltration (Millipore). The concentrated sample was ultracentrifuged at 75,325$g$ for 15 min and applied to a Superdex 200 Increase column (GE Healthcare) equilibrated with GF buffer (0.03% GDN, 20 mM Tris-HCl pH 8.0 and 150 mM NaCl). The peak fractions were combined and concentrated down to 30 µl using Ultrafree 100-kDa MWCO ultrafiltration. Purity was checked by SDS–PAGE (Extended Data Fig. 6a). The final protein concentration was approximately 10 mg ml⁻¹.

### Grid preparation

The purified complex was split into two. The first half was used to obtain structures in acidic condition. Then, 4 µl of protein was mixed with 1 µl of 50 mM citric acid buffer (the 50 mM citric acid buffer was prepared by diluting 0.5 M citric acid–sodium citrate buffer at pH 4.0) immediately before applying the sample to the grid. The pH after mixing was measured using a pH-indicator strip to be pH ~5.0–5.5. The time from mixing to freezing was less than 30 s. Next, 2 µl of protein solution was applied to an UltraAuFoil R1.2/1.3 (300 mesh) and plunged into liquid ethane using Vitrobot Mark4 (Thermo Fisher). The freezing parameters were as follows: blot force, 12; blot time, 4.5 s; temperature, 4 °C; humidity, 100%; wait time, 10 s; and drain time, 0 s. Filter paper was doubled to facilitate blotting. To prevent aggregation in acidic conditions, it was critical to use UltraAuFoil and to reduce the time between acidifying and freezing. Optimal freezing conditions were determined by inspecting the grids using Glacios (Thermo Fisher). The second half was frozen directly to prepare the vitrified grid without

any treatment at pH 8.0. We note that a Quantifoil R1.2/1.3 (300 mesh, Cu–C membrane) was used for pH 8.0, as the choice of Quantifoil over UltraAuFoil had no effect on the conformation of the NTDs at pH 8.0 (ref. 70) and the beam alignment was simpler to monitor during the EPU session with Quantifoil carbon membrane grids.

### Cryo-EM imaging

All data were collected using a Titan Krios G4i (Thermo Fisher) equipped with a BioQuantum K3 detector at Vanderbilt University's cryo-EM facility. Images were collected at 50 frames per video. The aberration-free image shift function was used in EPU (Thermo Fisher) semiautomated data collection software. The microscope was equipped with fringe-free optics, which enabled a smaller beam diameter for imaging. An objective aperture was not used. The detector dose rate was at 15.6–15.7 e⁻ per pixel per s (measured over ice). The total dose was at 52.8–55.6 e⁻ per Å² (measured over vacuum). Each video contained 50 frames. The detailed parameters used for data collection in each sample are summarized in Table 2. Data collection was completed in a single EPU session for the sample at pH 8.0 (21,898 videos) but in two sessions (10,240 + 9,444 = 19,684 videos) for the sample at pH 5.5. Representative motion-corrected images are shown in Extended Data Fig. 6d–f.

### Image processing of cryo-EM data

All image processing was performed using RELION 4 and 5 (refs. 71,72). Each raw video stack (50 frames) was motion-corrected (at 4 × 4 patches) and dose-weighted using MotionCor2 (ref. 73). CTFFIND4 was used to estimate the contrast transfer function (CTF) from non-dose-weighted images using 1,024 × 1,024 pixel tiles[74]. No symmetry was imposed throughout. Initial particles were identified using Autopick (pH 8.0, 7,593,346 particles; pH 5.5, 7,028,468 particles). Templates for Autopick containing 2D class averages of particles were centered at the gate region. Thus, using an optimal circular mask, the 2D and 3D classification was guided mostly from the signals in the LBD, TMD and TARPγ2. Before 2D classification, particles were extracted from a box size of 360 × 360 pixels and rescaled to 128 × 128 pixels. Parameters for 2D classification were the VDAM algorithm (variable-metric gradient descent algorithm with adaptive moments estimation) with 200 minibatches and regularization parameter $T = 2$. Mask circles were chosen at 180-Å diameter to purposefully cut off a portion of the NTDs, such that the alignment would be dominated by the signals in the LBD, TMD and TARPγ2. Particles belonging to 2D class averages with secondary-structure features of AMPAR were selected (pH 8.0, 1,886,582 particles; pH 5.5, 3,051,235 particles). The heterogeneity of the NTD layer was substantially different at the two pH values, even at the initial 2D class averages. The 2D class averages in the main figures with complete NTDs were produced by re-extracting the aligned particles by recentering them to ensure the entire architecture was contained in the circular mask.

Before 3D classification, particles were extracted from a box size of 360 × 360 pixels and rescaled to 180 × 180 pixels to optimize computational load. The 3D classification was performed for 40 iterations at $T = 4$ without a mask and using EMD-29386 (GluA2*flip*(Q) in complex with TARPγ2(KKEE)) as the initial model[50]. We also conducted 3D classification using masks that incorporated conformational heterogeneity of the NTDs but the outcome was not substantially different, which confirms that the alignment at this stage was guided mainly from the signals in the LBD, TMD and TARPγ2. Four and six classes were specified for pH 8.0 and pH 5.5, respectively. Well-defined classes with clear features of transmembrane helices of GluA2 and TARPγ2 were selected (class 1 and 3 for pH 8.0; class 6 for pH 5.5). The particles in two classes were combined in pH 8.0. The numbers of particles selected after 3D classification were 1,108,462 particles (pH 8.0) and 813,615 particles (pH 5.5). Particles were then re-extracted from a box size of 360 × 360 pixels without binning. Further 3D refinement (Refine3D)

was performed using a mask that covered the LBD, TMD and TARPγ2 (LBD–TMD mask). The 3D refinement at this stage was performed using RELION 4 with the '--external_reconstruct' flag and SIDESPLITTER[75]. The 3D refinement was followed by postprocessing, which produced maps of around 2.8-Å overall resolution in both pH conditions. The maps were further improved by CTF refinement, followed by another iteration of Refine3D and postprocessing, which produced a consensus map of the LBD, TMD and TARPγ2 at an overall resolution of 2.8 Å (Table 2 and Extended Data Fig. 6e). Focused refinement of the NTD at pH 8.0 was conducted by first recentering the NTD and then refining using an NTD mask (the consensus NTD map is deposited as an associated map of EMD-44232). In the consensus reconstruction, the LBD adopted greater heterogeneity in the acidic conditions, noticeable as ill-defined LBDs. The local resolution, calculated by ResMap[76], of the LBD in the consensus map of the D1 lobe that is closer to the NTD was much lower at pH 5.5 (Extended Data Fig. 6f). The alignment was guided toward improving the resolution of the membrane-embedded region at the cost of degrading the alignment of the LBD because the former contains many bundles of α-helices that generate strong signals.

To resolve the heterogeneity of the LBD and the NTD, the particles from the consensus alignment were re-extracted from a box size of 360 × 360 pixels and rescaled to 128 × 128 pixels, preserving the alignment parameters, and refined using the LBD–TMD mask with a local search. For each pH condition, a mask that covered the NTD and LBD was generated. To generate the mask, one round of 3D classification without alignment was conducted without a mask to sample the conformational heterogeneity of the NTDs at each pH. Representative classes that defined the range of heterogeneity were added in Chimera to guide mask production. Next, the signals outside the NTD and LBD were subtracted using the above mask and the particles were classified into 20 and 40 classes without alignment using regularization $T = 4$ for 40 iterations (representative classes are shown in Extended Data Fig. 7b,c). The extent of heterogeneity was low at pH 8.0 and, thus, classification into 20 classes was sufficient to sample the entire range of heterogeneity because similar conformations were present among classes. In contrast, classification into 40 classes produced a variety of splayed NTD conformations with unique NTD 3D arrangements. At pH 8.0, classes 1 (containing 47,782 particles) and 12 (containing 48,399 particles) were chosen as representative classes for further refinement of the full complexes. At pH 5.5, class 23 (containing 29,945 particles; Fig. 5c) was the only class that contained solid signals of both NTD dimers and was, thus, subjected to further refinement of the full complex. The reason for the conformational stability in class 23 is because of both NTDs approaching the LBDs, possibly making weak contacts. We note that, because of extensive conformational heterogeneity, one of the two NTD dimers was always weaker in other classes at pH 5.5, which prevented 3D reconstruction of the full complex. To produce the full map of above classes, focused refinement was conducted using the LBD–TMD mask to obtain the maps containing the LBD, TMD and TARPγ2, whereas the particles were recentered to the NTD layer by shifting the center by 64.8 Å in the z direction and refined using the NTD mask to obtain the NTD maps. The NTD layer in class 23 at pH 5.5 still contained conformational heterogeneity that prevented high-resolution 3D reconstruction, which resulted in an overall resolution of 5.9 Å. All other maps of subclassified conformations were refined to a final resolution ranging from 3.4 to 3.7 Å (Table 2).

To further understand the LBD–TMD conformations relative to the NTD conformations, the LBD, TMD and TARPγ2 portions of various classes were investigated by focused refinement using the LBD–TMD mask. In addition to class 23 introduced above, classes 8, 9, 16, 18, 19, 20, 29, 31, 37 and 40 were chosen as representative splayed NTD conformations at pH 5.5. Similarly, in addition to classes 1 and 12 introduced above, classes 4, 6, 8 and 15 were chosen as representative compact NTD conformations at pH 8.0. For each pH condition, a mask that covered the NTD and LBD was generated. Particles in each class were re-extracted

from a box size of 360 × 360 pixels and rescaled to 180 × 180 pixels. Each class was subjected to focused refinement using the LBD–TMD mask. Refine3D and postprocessing produced maps at overall resolution ranging from 3.7 to 4.0 Å at pH 5.5 and from 3.4 to 3.9 Å at pH 8.0. The differences between the LBD conformations were characterized as translation and rotation between the two LBD dimers, which were small conformational differences in the organization of the LBDs in the gating ring[55]. The overall resolutions of the maps were estimated using a Fourier shell correlation (FSC) = 0.143 cutoff in RELION[77]. The image processing and model statistics are summarized in Table 2. Angular distributions of assigned angles were inspected to ensure the coverage of the Fourier space. Visual inspection of the map showed no signs of artifacts.

## Model building and refinement

The model building and refinement for the consensus maps, maps of class 1 and 12 at pH 8.0 and the map of class 23 at pH 5.5 (Fig. 5c) were conducted as follows: the reference models, PDB 8FPG (TMD and TARPγ2) and PDB 8FPK (LBD), were rigid-body fit into the EM density map using Chimera[78]. The fit was further adjusted using the jiggle fit function in Coot[79]. Further manual adjustment with the real-space refine zone function in Coot was used to generate an atomic model. The generated model was further refined using the real_space_refine tool in Phenix[80]. Real-space refinement was conducted by imposing secondary-structure restraints by annotating helices and sheets in the PDB file. To prevent overfitting of the models into the density, refinement was run for five cycles with strict geometric restraints of 0.005–0.01 for bond length and 0.5–1 for bond angle. MolProbity and Mtriage were used for validation. To interpret the full architecture, the map produced by Refine3D, the unmasked and unsharpened map, was used to position the NTD map using rigid-body fit in Chimera. The maps of classes 4, 6, 8 and 15 at pH 8.0 and classes 8, 9, 16, 18, 19, 20, 29, 31, 37 and 40 at pH 5.5 were interpreted as follows: The maps were thresholded at an optimal level to visualize the densities of the NTDs. The NTD atomic model from class 12 at pH 8.0 was rigid-body fit into the map using Chimera. No further model refinement was conducted for the NTDs. The atomic model of the consensus map was rigid-body fit into the LBD, TMD and TARPγ2 densities using Chimera. The model was subjected to jiggle fit and all-atom refinement in Coot with the Geman–McClure self-restraint at 4.2 Å (refs. 27,81). PyMOL (Schrödinger) and Chimera were used to further analyze the structure and generate figures.

## Size-exclusion chromatography with MALS

The molecular mass of NTDA2 was determined in solution using size-exclusion chromatography (SEC) with MALS (SEC–MALS). Measurements were performed using a Wyatt Heleos II 18 angle light scattering instrument coupled to a Wyatt Optilab rEX online refractive index detector. Samples of 100 μl were resolved in 10 mM HEPES and 150 mM KCl (pH 7.5) buffer on a Superdex S200 10/300 analytical gel filtration column coupled to an Agilent 1200 series liquid chromatography system running at 0.5 ml min⁻¹ before then passing through the light scattering and refractive index detectors in a standard SEC–MALS format. Protein concentration was determined from the excess differential refractive index based on 0.186 ΔRI for 1 g ml⁻¹. The measured protein concentration and scattering intensity were used to calculate the molecular mass from the intercept of a Debye plot using Zimm's model as implemented in the Wyatt ASTRA software.

The experimental setup was verified using a BSA standard run of the same sample volume. The monomer peak was used to check mass determination and to evaluate interdetector delay volumes and band-broadening parameters that were subsequently applied during the analysis of NTDA2.

## DNA constructs and culture for electrophysiology

Sequences for rat GluA2 and rat GluA1 were flip variants; GluA2 was unedited at the 586Q/R site and edited at the 743R/G site. All cDNA

constructs used for transfection were generated using in vivo assembly cloning as previously described[82]. Constructs were cloned in either pRK5 or IRES vectors. GluA2$_{delNTD}$ was made by deleting the NTD sequence from position 1 to 394 (mature peptide). GluA1$_{NTDA2}$ was made by replacing the NTD from A1 (mature peptide residues from 1 to 390) with the residues from A2. Similarly, for GluA2$_{NTDA1}$, we replaced the NTD of GluA2 (mature peptide, residues 1–394) with the corresponding sequence from A1.

HEK293T cells (American Type Culture Collection, cat. no. CRL-11268, RRID: CVCL_1926, lot 58483269; identity authenticated by short tandem repeat analysis, *Mycoplasma* negative), cultured at 37 °C and 5% $CO_2$ in DMEM (Gibco; high-glucose, GlutaMAX, pyruvate, cat. no. 10569010) supplemented with 10% FBS (Gibco) and penicillin–streptomycin, were transfected using Effectene (Qiagen) according to the manufacturer's protocol. Transfected cells were identified by cotransfection of a pN1-EGFP plasmid or by EGFP–mCherry coexpressed from the pIRES2 plasmid. Where cotransfected, the DNA ratio of AMPAR to TARPγ2 was 1:2. To avoid AMPAR-mediated toxicity, 30 µM NBQX (Tocris or HelloBio) was added to the medium immediately after transfection.

### Electrophysiology

Recording pipettes were pulled with a P1000 horizontal puller (Sutter Instruments) using borosilicate glass electrodes (1.5 mm outside diameter, 0.86 mm inside diameter; Science Products). Electrode tips were heat-polished with an MF-830 microforge (Narishige) to final resistances of 2–4 MΩ (whole cell) and 6–10 MΩ (outside-out patches). Electrodes were filled with an internal solution containing (in mM) CsF (120), CsCl (10), EGTA (10), HEPES (10), Na2-ATP (2) and spermine (0.1), adjusted to pH 7.3 with CsOH. The extracellular solution contained (in mM) NaCl (145), KCl (3), CaCl2 (2), MgCl2 (1), glucose (10) and HEPES (10), adjusted to pH 7.4 using NaOH. We used 1 M HCl to adjust the pH to 5.5 and 6.4 for the recordings in acidic conditions. Recordings were performed at room temperature (-21–23 °C). Currents were recorded with an Axopatch 700B amplifier (Molecular Devices) 24–48 h after transfection. Signals were prefiltered at 10 kHz with a four-pole Bessel filter, sampled at 100 kHz with the Digidata 1550B (Molecular Devices), stored on a computer hard drive and analyzed using pClamp 10 (Molecular Devices), Excel and GraphPrism software.

On the day of recording, cells were plated on poly(L-lysine)-treated glass coverslips. Fast perfusion experiments were performed with a two-barrel theta tube glass with a diameter of approximately 250 µm. The theta tube was mounted on a piezoelectric translator (Physik Instrumente) and the command voltage (9 V) was filtered with a 500-Hz Bessel filter to reduce mechanical oscillations. The theta tube was filled with pressure-driven solutions (ALA Scientific Instruments). The speed of solution exchange at the theta tube interface was measured as 20–80% of the rise time of the current generated with 50% diluted extracellular solution and was on average about 120 µs (outside-out patches) or 400 µs (whole cell). Patches were voltage-clamped at −60 mV (voltage not corrected for junction potential of 8.5 mV). Series resistance was not corrected for outside-out recordings. For whole-cell recordings, series resistance was never higher than 8 MΩ and was compensated by 90%.

Recovery from desensitization was measured with a two-pulse protocol. A conditioning pulse of 10 mM glutamate with a duration of 200 ms or 100 ms (Fig. 1e) was followed by 15-ms glutamate pulses delivered at intervals increasing by 5 or 10 ms (GluA2 constructs) or 20 ms (GluA1 wt and GluA1$_{NTD\_A2}$ constructs). Desensitization (200-ms glutamate pulses) and deactivation (1-ms glutamate pulses) time constants were obtained by fitting the current decay (Chebyshev algorithm, built-in Clampfit 10.2; Molecular Devices) of the glutamate application from 90% of the peak to the steady-state or baseline current with one or two exponentials. Where biexponential fits were used, the weighted $\tau_{des}$ is reported, calculated as follows: $\tau_{w,des} = \tau_f(A_f/(A_f + A_s)) + \tau_s(A_s/(A_f + A_s))$, where $\tau_{f/s}$ and $A_{f/s}$ represent the fast/slow component time constant

and coefficient, respectively. The rise time constant was obtained by fitting the current rising phase (from 1-ms glutamate application) with one exponential from 20% to the peak.

Recovery from desensitization was fitted by a Hodgkin–Huxley-type equation:

$$f(t) = y_0 + (y_{max} - y_0) \times (1 - \exp(-kt))^m$$

where $y_0$ and $y_{max}$ are the minimum and maximum, $k$ is the rate constant, $t$ is the interpulse interval and $m$ is the slope. GluA2 receptors have a steeper recovery profile; therefore, we fixed the slope to 2 (ref. 83). Recovery profiles of GluA1 are much slower than GluA2; therefore, the recovery time constant for GluA1 receptors was obtained with $m = 1$, which gives a single exponential function.

The dose–response relationship of GluA2 + TARPγ2 at pH 7.4 and 5.5 was measured from whole-cell currents at a holding voltage of −40 mV. Six concentrations of glutamate were applied with a theta tube to a lifted whole cell to obtain the dose–response relationship.

The dose–response relationship for each cell was fitted with GraphPrism software using the Hill equation:

$$I = \frac{I_{max}[A]^{nH}}{[A]^{nH} + EC_{50}^{nH}}$$

where $I_{max}$ is the maximum response, $EC_{50}$ is the concentration of glutamate that gave half of the maximum response and $n_H$ is the Hill coefficient.

For the presentation, dose–response relationships from each cell were normalized to the response of 10 mM glutamate and pooled together (Extended Data Fig. 3d).

Nonstationary fluctuation analysis (NSFA) was performed on the desensitizing current phase of macroscopic currents evoked with glutamate pulses (10 mM, 200 ms) from outside-out patches containing GluA2 + TARPγ2. The same patch was exposed to pH 7.4 and 5.5 and at least 30–100 successive responses were collected for each condition from the same patch. The mean current and variance from successive responses were calculated in Clampfit and imported to a custom-written Python script, where the variance $\sigma^2$ was grouped in ten amplitude bins, plotted against the mean current and fitted with a parabolic function[38]:

$$\sigma^2 = i\bar{I} - \frac{\bar{I}^2}{N} + \sigma_B^2$$

where $i$ is the single-channel current, $\bar{I}$ is the mean current, $N$ is the number of channels and $\sigma_B^2$ is the background variance. The weighted mean single-channel conductance $\gamma$ was obtained from the single-channel current and the holding potential (−60 mV, not corrected for the liquid junction potential).

Data visualization and statistical analysis were performed using GraphPad Prism.

### AlphaFold and energetic modeling

Predicted structural models of the homomeric and heteromeric GluA1 NTD (UniProt P19490; residues 19–400) and GluA2 NTD (UniProt P19491; residues 25–398) were generated using AlphaFold2-Multimer[47] through ColabFold[84]. The highest-ranked predictions were all validated against predefined established criteria (PAE, pTM, pLDDT, DockQ, Mol-Probity and QS-score). The Dynamut2 server[48] was used to investigate missense substitutions on the GluA2 BD NTD interface using a model from the previously published GluA2/A1 complex[52].

### Dissociated hippocampal cultures

All procedures were carried out under PPL 70/8135 in accordance with UK Home Office regulations. Experiments were licensed under the

UK Animals (Scientific Procedures) Act of 1986 following local ethical approval. All animals were housed with food and water ad libitum on a 12-h light–dark cycle at room temperature (20–22 °C) and 45–65% humidity.

Cultures were prepared according to the protocol described in Beaudoin et al.[85]. Hippocampi from postnatal P0–P1 C57BL/6JOla wt mice were dissected in ice-cold HBSS (Ca$^{2+}$ and Mg$^{2+}$ free; Gibco, cat. no. 14175095) containing 0.11 mg ml$^{-1}$ sodium pyruvate (Gibco, cat. no. 12539059), 0.1% glucose and 10 mM HEPES (Gibco, cat. no. 15630056) and dissociated for 20 min at 37 °C with trypsin (0.25% w/v; Gibco, cat. no. 15090-046). Neurons were plated onto glass coverslips (24-mm round coverslips 1.5; Glaswarenfabrik Karl Hecht, cat. no. 1001/24_15 92100105080) coated with poly(L-lysine) (0.1 mg ml$^{-1}$; P2636, Sigma-Aldrich) following resuspension in equilibrated plating medium containing 86.55% MEM (Gibco, cat. no. 21090022), 10% heat-inactivated FBS (Gibco, cat. no. 11573397), 0.45% glucose, 1 mM sodium pyruvate and 2 mM GlutaMax (Gibco, cat. no. 35050038). Cultures were kept at 37 °C and 5% $CO_2$ in equilibrated maintenance medium containing 96% Neurobasal plus medium (Thermo Fisher, cat. no. A3582901), 1× B-27 plus supplement (Thermo Fisher, cat. no. A3582801) and 2 mM GlutaMax. Half of the medium was replaced every 3–5 days.

### FRAP

Dissociated hippocampal neurons were made to express SEP-tagged AMPAR constructs using either Lipofectamine 3000 (Thermo Fisher) at 11 days in vitro and imaged at 14 days in vitro. SEP-tagged GluA2 (SEP–GluA2) was created by inserting the fluorescent protein-coding region between the third and fourth residues of the mature GluA2 protein. In addition, a SEP–GluA2 F231A mutant was generated. In all constructs, the SEP tag was preceded and followed by an A-S dipeptide linker. The SEP sequence was kindly provided by J. Hanley.

At 14–15 days in vitro, neurons were imaged in artificial cerebrospinal fluid containing (in mM) NaCl (150), KCl (2.5), MgCl$_2$ (2), CaCl$_2$ (2), HEPES (20) and glucose (10) at pH adjusted to 7.4 or 5.5 with either NaOH or HCl in a heated chamber at 37 °C. Images were acquired on a Zeiss 780 laser scanning confocal microscope using a ×40 (1.2 numerical aperture) water-immersion objective with a pixel size of 100 nm. Photobleaching was achieved by repeated $xy$ scanning of the region of interest (2 μm$^2$) at high laser intensity, using excitation at 405 nm for Fig. 6c. The imaging protocol consisted of 3 images and 20 images taken before and after bleaching, respectively, at 30-s intervals. Analysis was performed using EasyFRAP-web[86]. Photobleaching because of image acquisition was corrected by normalization to the fluorescence of the distant nonphotobleached spine (2 μm$^2$) and to the background fluorescence. Normalized data were further postprocessed and fitted to a single exponential curve using GraphPad Prism.

### MD

**GluA2 NTD MD simulations.** We used the highest-resolution GluA2 homodimeric NTD crystal structure obtained at pH 8.0 (PDB 3H5V; 2.33 Å)[42] as the starting model for NTD-only constant-pH simulations. To prepare the physiological homotetramer (dimer of dimers) from the deposited trimeric asymmetric unit (ASU), a copy of the unit was rotated 180° around the vertical axis (perpendicular to the membrane in the full-length GluA2 receptor), superimposed onto the unrotated ASU with Chimera MatchMaker[78] and had repeated domains removed. The created NTD homotetramer was validated against NTD domains of full-length GluA2 crystal structures with a QMEANDisCo score (SWISS-MODEL QMEAN webserver, version 3.1.0)[87,88] of 0.87 ± 0.05 (score range 0–1; scores > 0.6 indicate good agreement with experimental structures).

Fixed protonation states were assigned to all titratable residues with the Henderson–Hasselbalch equation on the basis of the pH being simulated and pKa calculations using PROPKA3 (version 3.4.0)[89,90]; pKa values were calculated on NTD homotetrameric (created as above) crystal structures obtained at pH 8.0 (PDB 3H5V)[42] and pH 4.8 (PDB 3HSY)[18] for basic and acidic simulations, respectively. By assigning acidic protonation states to the basic pH starting model, we simulated an instantaneous change in pH at the start of the acidic simulations. The NTD tetramer was placed at the center of a 175 × 175 × 175 Å$^3$ box with periodic boundary (PB) conditions (buffer distance between the protein and PB set to 1.0 nm), solvated with simple point-charge water[91] and charge-neutralized with sodium and chloride ions. This resulted in a system with 544,812 atoms.

All simulations were run in GROMACS (version 2019.3)[92–94]. The system was first energy-minimized over 10,000 steps with a step size of 0.01, followed by three 1-ns equilibrations with 2-fs steps: temperature (NVT) equilibration to 300 K with the v-rescale thermostat, first pressure (NPT) equilibration to 1 bar with the Berendsen barostat and second NPT equilibration to 1 bar with the isotropic Parrinello–Rahman barostat for greater accuracy; temperature was controlled for the protein and solvent groups separately. The protein movement was fully constrained during the first two equilibrations to not destabilize the system. Finally, the system was simulated with the v-rescale thermostat and Parrinello–Rahman barostat for 100 ns. All simulations were run with the Verlet cutoff scheme, LINCS H-bond constraints, a 1.2-nm van der Waals cutoff and particle mesh Ewald electrostatics.

### Reporting summary

Further information on research design is available in the Nature Portfolio Reporting Summary linked to this article.

### Data availability

Cryo-EM coordinates and corresponding EM maps were deposited to the PDB and EM Data Bank under the following accession codes: PDB 9B5Z (EMD-44232), PDB 9B60 (EMD-44233), PDB 9B67 (EMD-44248), PDB 9B68 (EMD-44249), PDB 9B6A (EMD-44251), PDB 9B69 (EMD-44250), PDB 9B61 (EMD-44234), PDB 9B63 (EMD-44244) and PDB 9B64 (EMD-44245). MD simulation trajectories were deposited to Zenodo (https://doi.org/10.5281/zenodo.11654387)[95]. Requests for materials (plasmids and cell lines) will be fulfilled for reasonable inquiries and should be addressed to I.H.G. and T.N. Source data are provided with this paper.

### Code availability

The following software was used for MD simulations: SWISS-MODEL QMEAN webserver version 3.1.0 (open source), Modeller version 9.22 (open source), PropKa version 3.1 (open source), GROMACS version 2019.3 (open source), CHARMM-GUI webserver version 2019 (open source), PDB 2PQR webserver version 2.1.1 (open source) and CONAN version 2018 (open source).

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

## Acknowledgements

We thank C. Johnson for running the MALS experiment, N. Barry and J. Boulanger for helpful comments on FRAP imaging and analysis, R. Lape for providing the NSFA script and J.-N. Dohrke for helpful suggestions concerning the MD simulations. We also thank the Greger lab, J. Krieger and J. Watson for comments on the paper. We acknowledge the technical support provided by the Laboratory of Molecular Biology (LMB) Biophysics Facility, the Ares Biomedical Facility, LMB scientific computing and the LMB EM Facility. We acknowledge the use of the cryo-EM facility at the Center for Structural Biology (maintained by M. Chambers, S. Collier and M. Haider), ACCRE graphics processing unit nodes (supported by National Institutes of Health grant 1S10OD032234-01) and the Distributed Online Research Storage core at Vanderbilt University. We thank K. Kim and P. Christov at the Vanderbilt Chemical Synthesis Core for synthesizing chemicals. This work was supported by grants from the Medical Research Council (MC_U105174197), the Biotechnology and Biological Sciences Research Council (BB/N002113/1) and the Wellcome Trust (223194/Z/21/Z) to I.H.G. and NIH grants (R56/R01MH123474 and S10OD030292-01) to T.N.

## Author contributions

I.H.G. conceptualized and supervised the study. I.H.G. wrote the paper with input from the coauthors. J.I. performed and analyzed the electrophysiological experiments with help from H.H. T.N. performed protein purification and cryo-EM data collection, EM data processing and model building. N.K. performed and analyzed the MD simulations. I.S. performed and analyzed the FRAP imaging data with help from V.K. A.M.S. performed AlphaFold modeling. Funding came from I.H.G. and T.N.

## Competing interests

The authors declare no competing interests.

## Additional information

**Extended data** is available for this paper at https://doi.org/10.1038/s41594-024-01369-5.

**Correspondence and requests for materials** should be addressed to Terunaga Nakagawa or Ingo H. Greger.

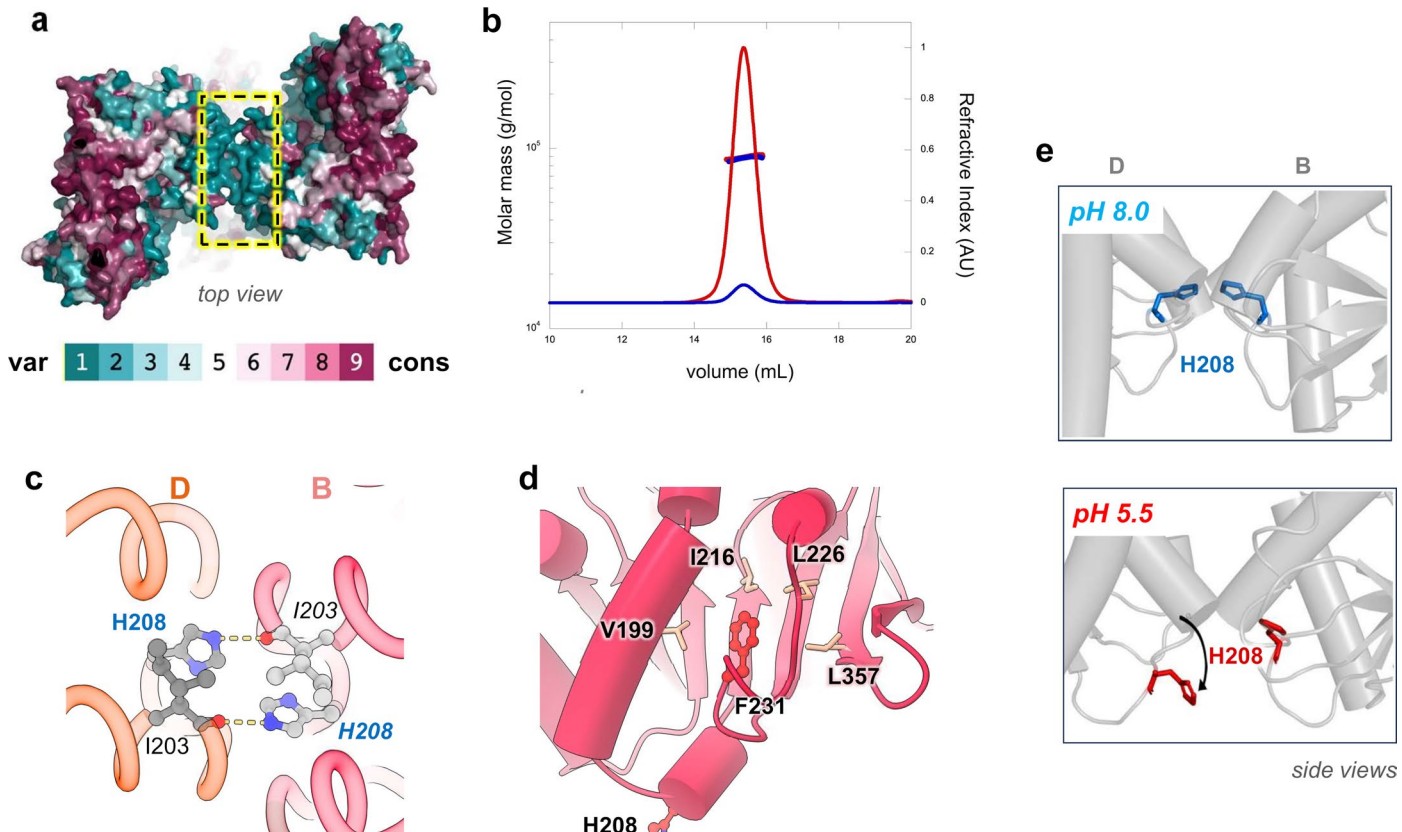

**Extended Data Fig. 1 | Features of the GluA2 BD NTD interface. a**, Conservation mapping using Consurf[96] illustrates high sequence variability of the BD NTD interface (boxed) between the four AMPAR paralogs (GluA1-4). The representation contains ~250 vertebrate AMPAR sequences. High sequence variability is shown in dark cyan ('var'), while conservation is shown in dark purple ('cons'). **b**, SEC-MALS chromatograms of excess refractive index for the GluA2 NTD injected at 13 mg/mL (red) and 1 mg/mL (blue). The number average mass across the peak as displayed was 89,000 and 88,000 g/mol respectively. **c**, Top view onto the BD interface from PDB 3H5V, highlighting the H-bonds between the H208 side chains and the I203 main chains. **d**, Zoom into the hydrophobic pocket holding the F231 side chain (PDB: 3HSY). **e**, All-atom MD simulation runs conducted at pH 8.0 (top) and at pH 5.5 (bottom). At acidic pH one of the histidines is pushed downwards (arrow) as a result of protonation and consequently charge repulsion. This will result in interface destabilisation.

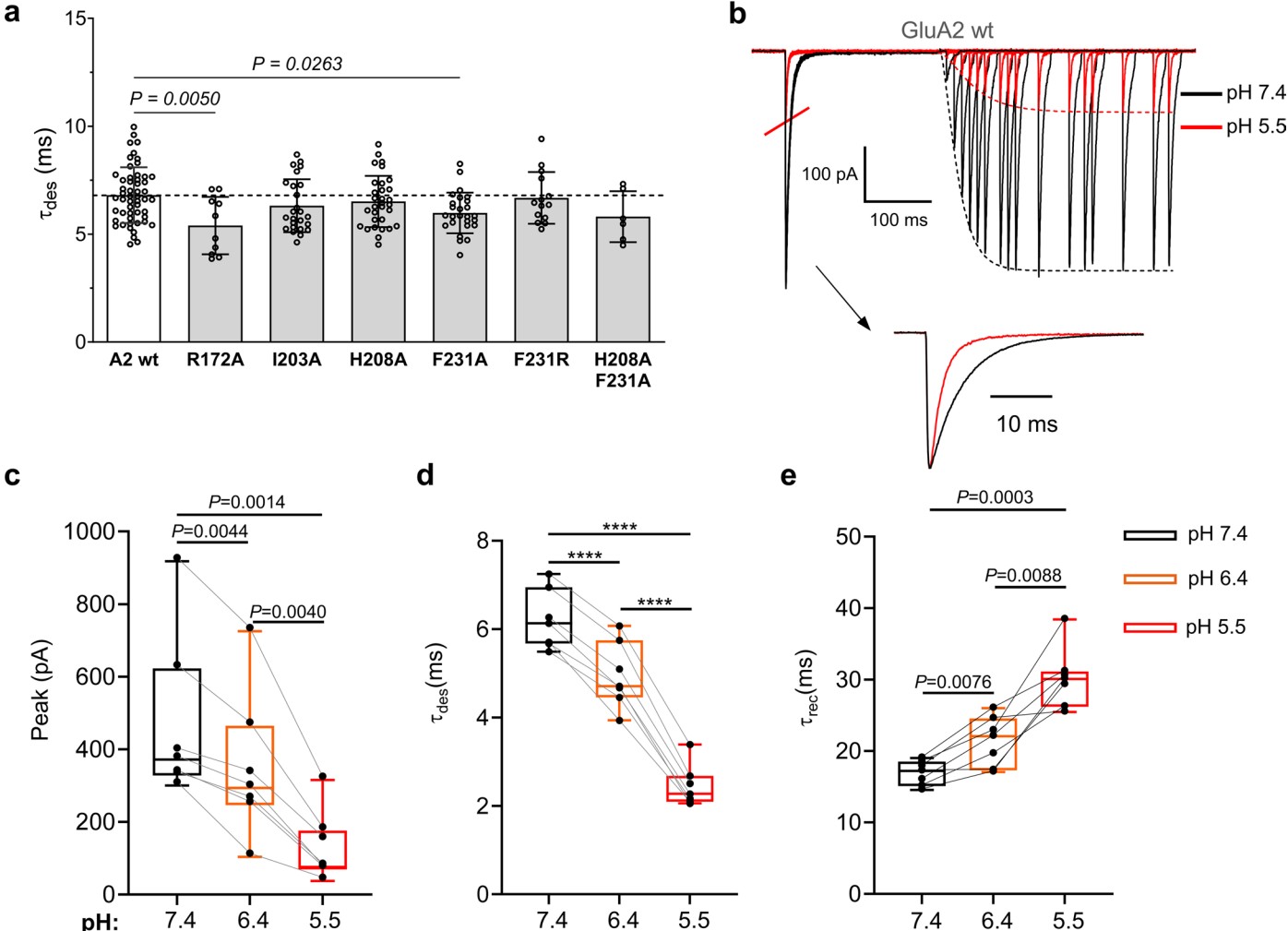

**Extended Data Fig. 2 | Recovery from desensitization is slowed by breaking the GluA2 interface by mutation or extracellular acidification.**
**a**, Desensitisation time constants for GluA2 mutants in NTD BD interface measured from patches from HEK cells expressing GluA2. Number of patches = 53, 10, 25, 31, 26, 14 and 5 for A2 wt, R172A, I203A, H208A, F231A, F231R and H208A/F231A respectively. Bars and errors as in Fig. 1e. One-way ANOVA $F_{(6, 158)}$ = 3.129, P = 0.0063, followed by Dunnett's multiple comparisons test. All differences between means with P < 0.05 are indicated. **b**, Representative example of paired-pulse current evoked by 10 mM L-glutamate from the excised membrane patch from a HEK293T cell expressing WT GluA2flipQ receptors. Black and red current traces represent current responses obtained in pH 7.4 and 5.5 from the same patch respectively. Bottom trace shows averaged and scaled to peak responses in pH 7.4 (black trace) and 5.5 (red trace). **c-e**, Boxplots

showing peak amplitude, entry, and recovery from desensitization measured in three different pH solutions; pH 7.4 (black), pH 6.4 (orange), and pH 5.5 (red). Responses were obtained from 200 ms applications of 10 mM L-glutamate. Lines connect the values obtained from the same patch. Boxes and whiskers as in Fig. 3b. The effect of pH on current peak amplitude was revealed by repeated measures ANOVA test $F_{(1.259, 7.557)}$ = 37.08, P = 0.0003, followed post hoc with Tukey's multiple comparisons test. P < 0.05 values are indicated. Similarly, the effect of pH on entry and recovery from desensitisation was indicated by repeated measures ANOVA test $F_{(1.688, 10.13)}$ = 502.2, P < 0.0001, followed post hoc with Tukey's multiple comparisons test: ****P < 0.001 (desensitisation entry). Repeated measures oneway ANOVA $F_{(1.313, 7.881)}$ = 40.78, P = 0.0001, followed post hoc with Tukey's multiple comparisons test, all differences between means with P < 0.05 are indicated (recovery from desensitisation).

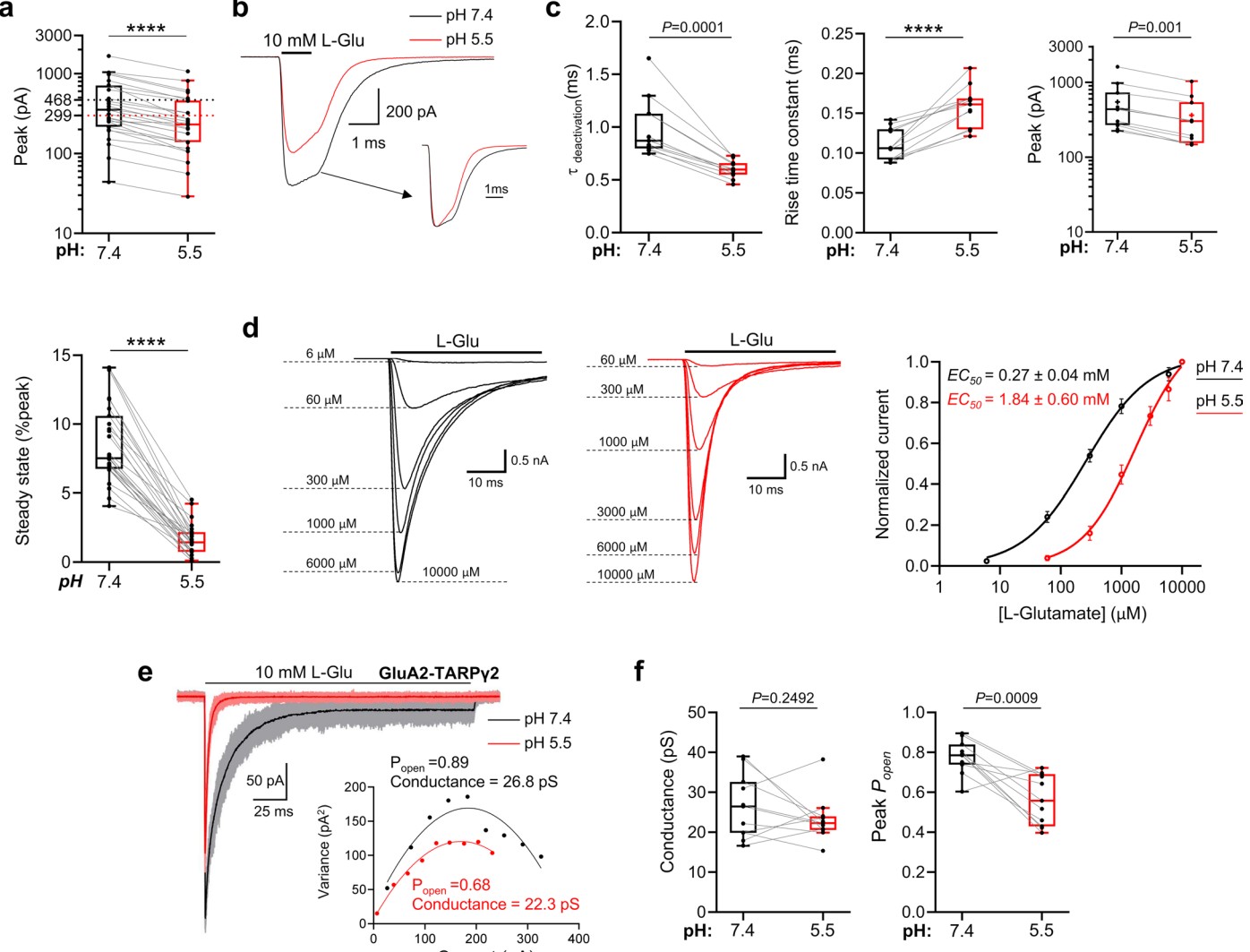

**Extended Data Fig. 3 | Acidic pH negatively modulates the kinetic of GluA2 + TARPγ2. a**, Peak amplitude (top panel) and steady state (bottom panel) measured from 200 ms of 10 mM L-glutamate applications in pH 7.4 (black) and 5.5 (red) from GluA2+TARP γ2, $n = 27$ patches. Boxes, whiskers, and lines as in Fig. 3b. Dotted lines show mean peak currents. Asterisks show comparisons for peak amplitude; two-sided, Wilcoxon matched-pairs signed rank test, $P < 0.0001$ and for steady-state; two-sided paired t-test: t = 14.98, df = 26, ****P = 0.0001. **b**, Representative responses evoked by 10 mM glutamate (1 ms, −60mV) in pH 7.4 (black line) and 5.5 (red line) from the outside-out patch (GluA2 + TARPγ2). The inset shows scaled-to-peak responses. **c**, Boxplots showing weighted deactivation time constant (left panel), rise time constant (middle panel), and peak amplitude (right panel) measured from current responses to 1 ms, 10 mM glutamate applications (GluA2 + TARPγ2, $n = 11$ patches). Boxes, whiskers and lines as in Fig. 3b. Asterisks show comparisons for deactivation time constant: two-sided pair sample t-test: $t = 6.167$, $df = 10$, ***P = 0.0001, rise time constant: two-sided pair sample t-test: $t = 6.574$, $df = 10$,****P < 0.0001 and peak amplitude: two-sided, Wilcoxon matched-pairs signed rank test, P = 0.001. **d**, Whole-cell

current responses (GluA2 + TARPγ2) to L-glutamate concentrations applied at pH 7.4 (black traces, left panel) and at pH 5.5 (red traces, middle panel). The left panel shows averaged and normalized concentration-response curves to L-glutamate obtained in pH 7.4 (black circles, $n = 6$ cells) and in 5.5 (red circles, $n = 6$ cells) error bars are SEM. Current responses were normalized to the response to 10 mM glutamate. **e**, Current responses from an outside-out patch containing GluA2 + TARPγ2 to 10 mM L-glutamate, 200 ms, holding voltage −60mV. Grey traces show responses in pH 7.4 and salmon traces show responses in pH 5.5. The black and red lines show averaged responses in pH 7.4 and 5.5. The inset shows the corresponding current-variance relationship, estimated channel conductance ($\gamma$), and open probability at the peak ($P_o$) in pH 7.4 (black) and 5.5 (red). **f**, Boxplots show the effect of pH 5.5 on the mean channel conductance ($\gamma$), and open probability estimates for the GluA2 + TARPγ2, $n = 11$ patches. Boxes, whiskers, and lines as in Fig. 3b. Indicated $P$ values are from the two-sided pair sample t-test, channel conductance: $t = 1.224$, $df = 10$, $P = 0.2492$ and the peak open probability: $t = 4.622$, $df = 10$, $P = 0.0009$.

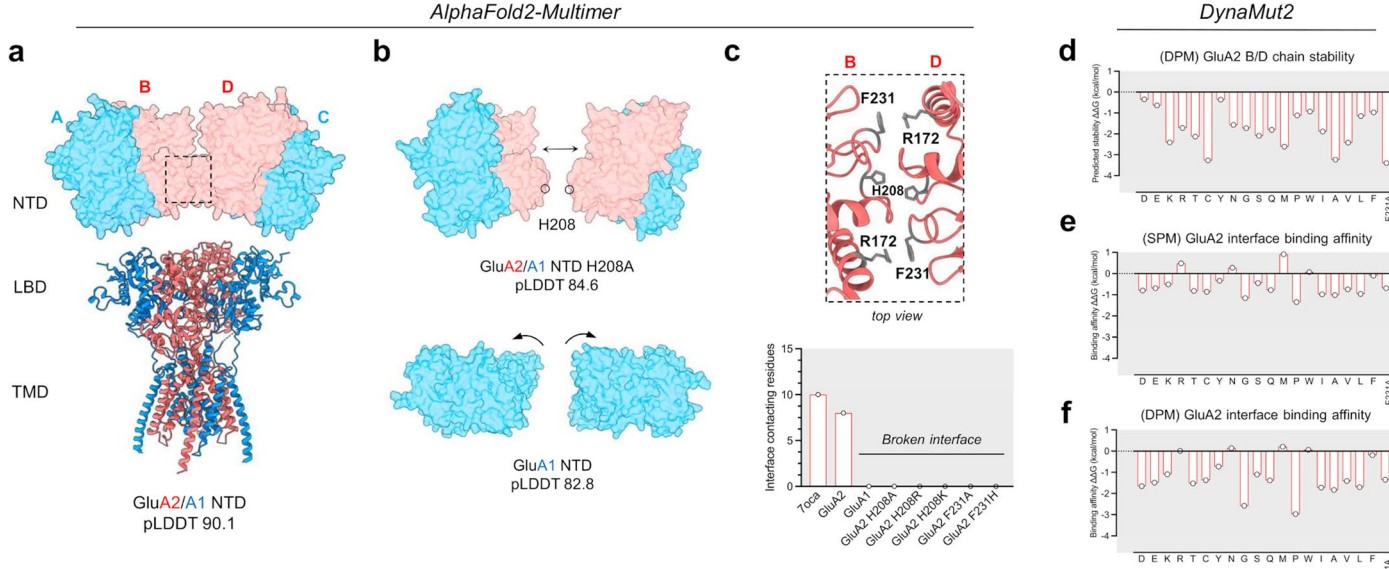

**Extended Data Fig. 4 | Modelling His208 as a key GluA2 NTD BD interface residue. a**, Structure of the GluA2 (red) and GluA1 (blue) hetero-tetrameric NTD by AlphaFold2-multimer predicts the favored occupation of GluA2 at the BD sites. Overlayed to the full-length AMPAR GluA1/GluA2 LBD and TMD (PDB 7oca). **b**, Structure of H208A mutated GluA2/A1 NTD by AlphaFold2-multimer predicts a breakage of the GluA2 BD interface (top), recapitulating the GluA1 homomer (bottom). **c**, Structure of the GluA2 (red) and GluA1 (blue) NTD by AlphaFold2-multimer accurately predicts key contacting residues (shown in grey) at the GluA2 BD interface. AlphaFold predictions with single point mutations at key

interface forming residues cause a complete disruption of the BD interface compared to WT models (GluA2) and previously published structures (PDB: 7oca). **d**, Double point missense mutations (DPM) at respective GluA2 BD chains at His208 are modelled to destabilize the NTD tier, calculated as the ΔGstability of the structure. **e–f**, Single (SPM) and double point missense mutations (DPM) at respective GluA2 BD chains at His208 are modelled to reduce the binding affinity between interface contacts in the NTD tier, calculated as the ΔGbinding affinity of the structure.

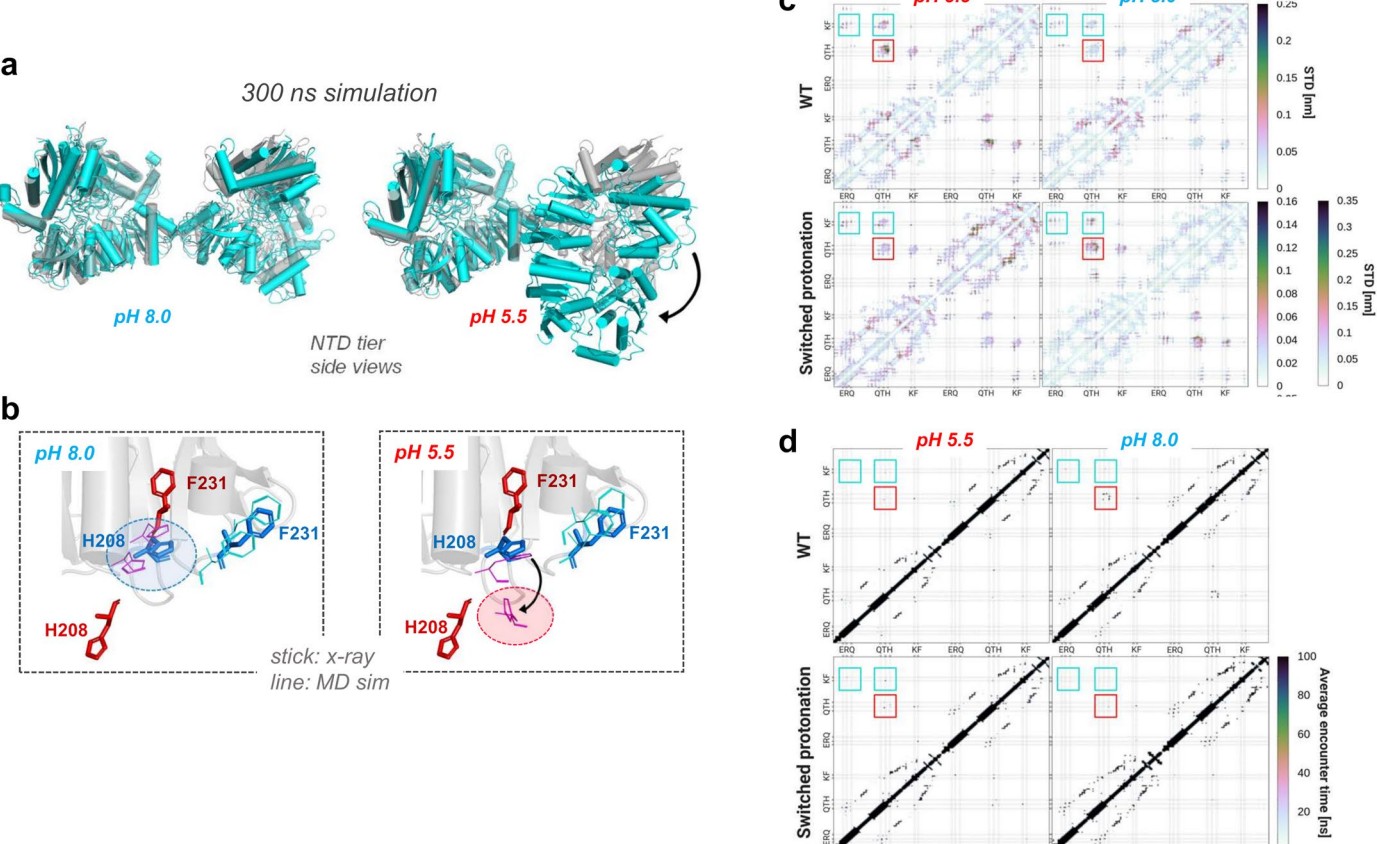

**Extended Data Fig. 5 | Behaviour of BD NTD interface residues in MD simulations. a**, Conformational changes between the NTD dimers seen after 300 ns at pH 5.5 (right panel) but not at pH 8.0 (left panel). **b**, Downward motion of the H208 side chain is seen in one run at pH 5.5 (red ellipsoid in right panel), approaching the x-ray reference structure (red sticks). This is not the case at pH 8.0 (blue ellipsoid in left panel). Position of the F231 residue is unchanged under both conditions. **c**, Residue interaction plots, showing standard deviations (STD; side bars) in pairwise residue distances. Top row: standard simulation, pH 5.5 left panel, pH 8.0 right panel. Bottom row: 'switched protonation' that is H208 is deprotonated in the pH 5.5 column and protonated in the pH 8.0 column. Boxed regions show changes in residue interactions (STDs) that are strongest between the H208 residues (QT<u>H</u>; red box) but are also seen between F231 (K<u>F</u>) and T204/H208 (QTH; right cyan box). This is not seen at pH 8.0 and is partly reversed in the 'switched protonation' panels. **d**, Interaction plots showing average encounter times between interface residues. Conditions were as described in the above panel c (see Methods for further details).

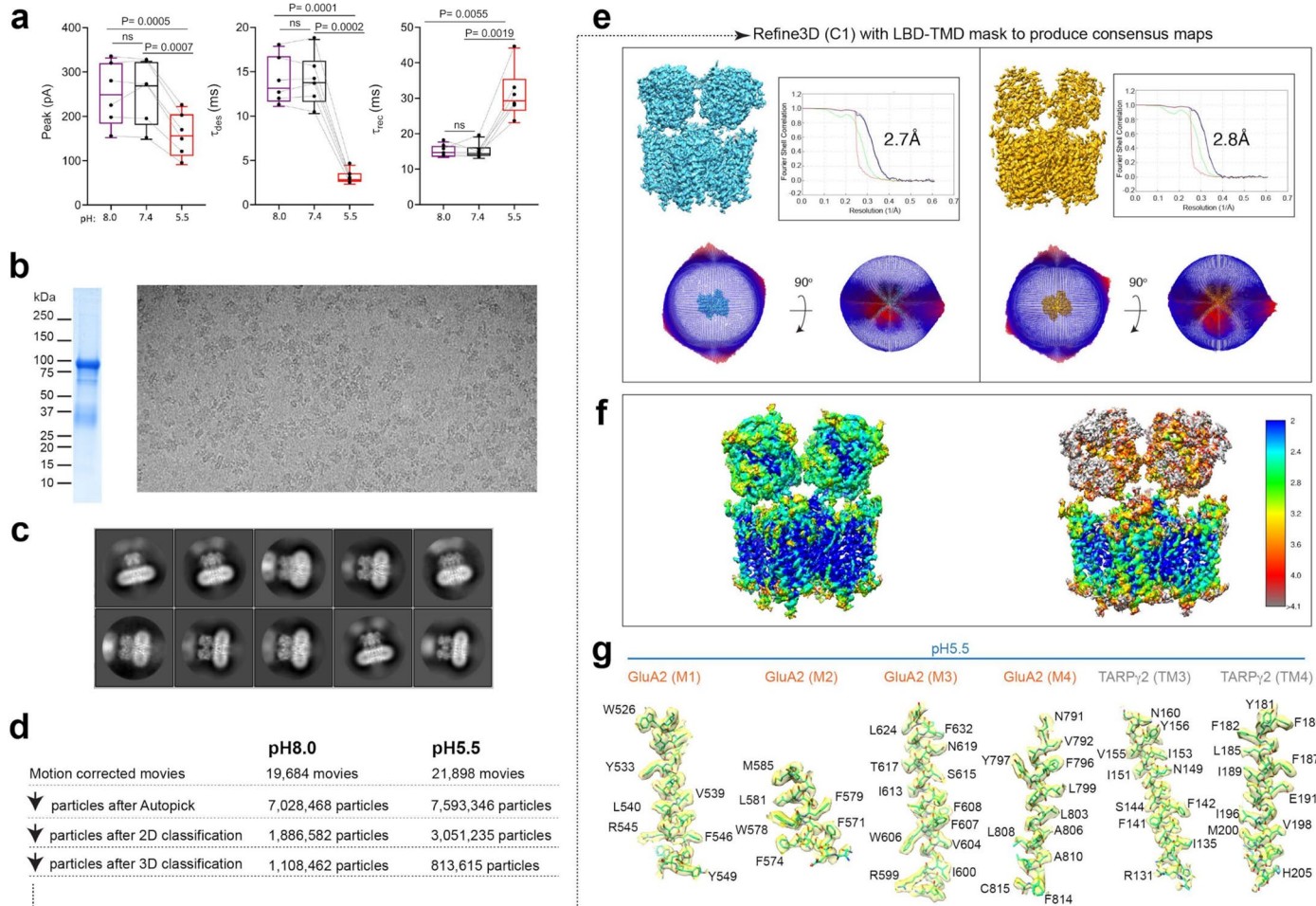

**Extended Data Fig. 6 | Cryo-EM data processing of GluA2flipQ/TARP γ2 complex at pH 8.0 and pH 5.5. a**, Peak amplitude, entry, and recovery from desensitization measured in pH 7.4 (black), pH 8 (purple), and pH 5.5 (red). Responses were obtained from 200 ms applications of 10 mM L-glutamate. Lines connect values obtained from the same patch. Boxes and whiskers as in Extended Data Fig. 2a. Effect of pH on current peak amplitude (left panel) was revealed by repeated measures ANOVA test $F_{(1.878, 9.392)}$ = 62.36, $P < 0.0001$, followed post hoc with Tukey's multiple comparisons test. Similarly, the effect of pH on entry (middle panel) and entry to desensitisation (right panel) was indicated by repeated measures one-way ANOVA $F_{(1.207, 6.036)}$ = 141.6, $P < 0.0001$, followed post hoc with Tukey's multiple comparisons test (desensitisation entry), repeated measures ANOVA test $F_{(1.116, 5.582)}$ = 37.06, $P = 0.001$, followed post hoc with Tukey's multiple comparisons test (desensitisation recovery). **b**, Left: 4–20% SDS-PAGE gel of the purified complex. GluA2flipQ migrating at 100 kDa and TARP γ2 at 37 kDa. Right: Representative (from 21,898 micrographs) motion-corrected cryo-EM image at pH 5.5. **c**, Representative 2D class averages (from 28 class averages containing clear secondary structural features) of particles at pH 5.5. Box size is 180×180 pixels, equivalent to 295.6 × 295.6 Å. **d**, Summary of particles that were selected sequentially through the image processing pipeline from the initial motion corrected micrographs. Selected particles were those that were contained in well-defined classes in each step. Particles that were selected after 3D classification were used for 3D refinement. **e**, Consensus maps calculated from focused refinement of the LDB, TMD, and TARP γ2 at pH 8.0 (left) and pH 5.5 (right). The Fourier Shell Correlation curves with estimated resolutions at FSC = 0.143 and angular distributions of the final particles are shown. Note the LBD densities at pH 5.5 are ill-defined compared to pH 8.0, due to conformational heterogeneity. **f**, Local resolutions of the maps containing the LDB, TMD, and TARP γ2 at pH 8.0 (left) and pH 5.5 (right), calculated using ResMap77. The heatmap reference is displayed at the right with local resolution in Å unit. **g**, Side chain resolution of the GluA2-TARPγ complex determined at pH 5.5; all GluA2 (M1-4) and TARP (TM1-4) are shown.

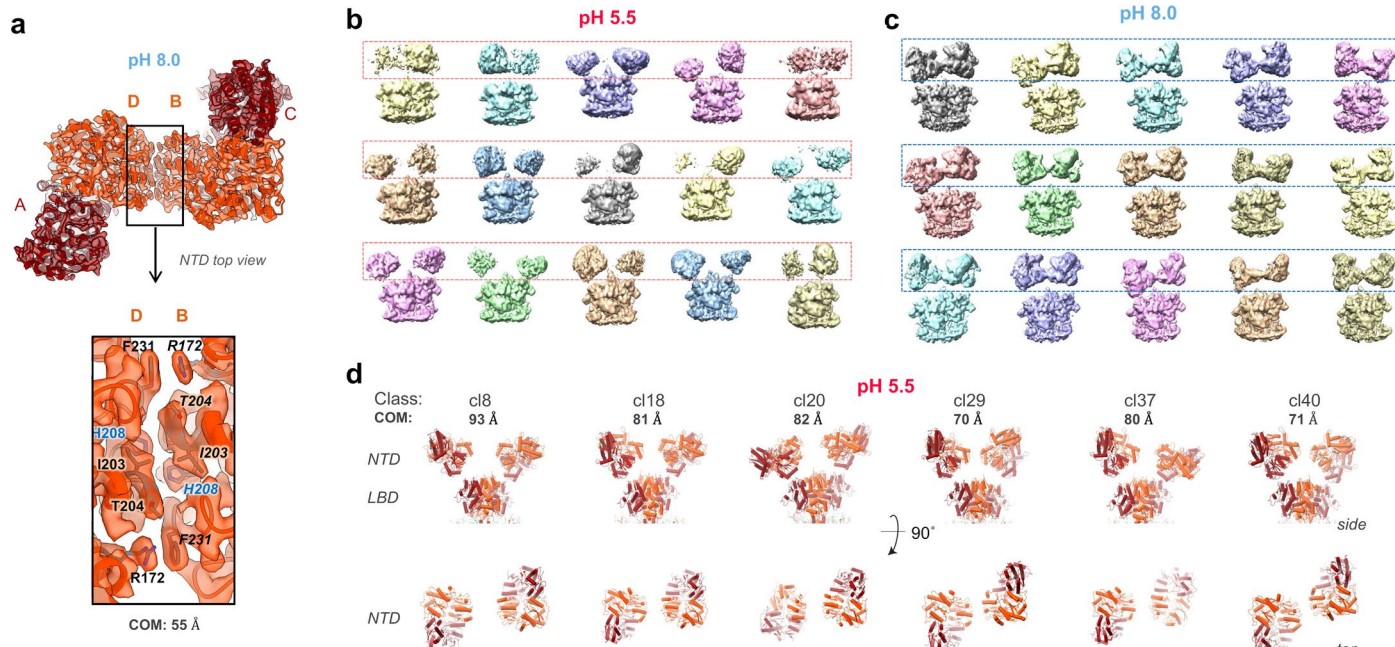

**Extended Data Fig. 7 | Conformations of the NTD tier at pH 8.0 and pH 5.5.**
**a**, 3D density map of the GluA2 NTD tier at pH 8.0. Upper panel: top view (BD interface boxed); lower panel: zoom into boxed region. **b,c**, Representative 3D classes of the NTD and LBD layers at pH 5.5 (**b**) and pH 8.0 (**c**). The NTD tiers are marked with dashed boxes. 15 representative classes out of 20 classes total. Particles within each class ranged from 34–90 K (pH 5.5), and 43–90 K (pH 8.0). **d**, Representative arrangements of the NTD tier at pH 5.5, shown as atomic models from two orthogonal views. Classes 8, 18, 20, 29, 37, and 40 were selected from extensive sub classification of particles into 40 classes (see Methods). The atomic models of the NTD and LBD were rigid-body fitted into the density map.

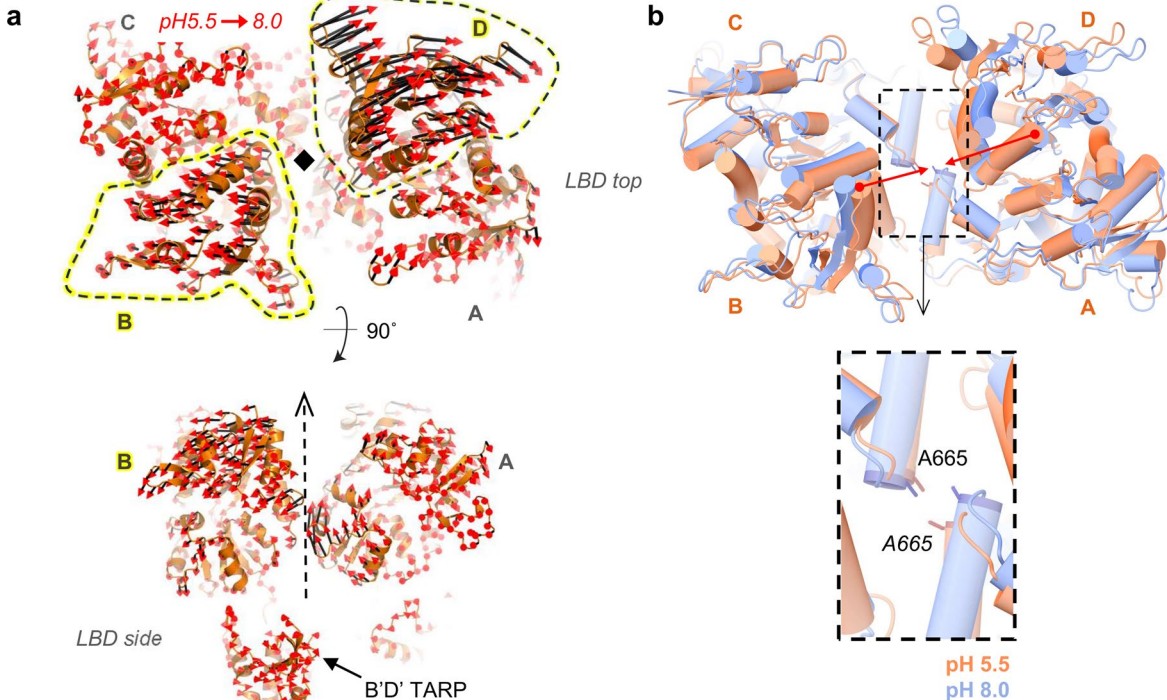

**Extended Data Fig. 8 | LBD rearrangements at pH 5.5 versus pH 8.0.**
**a**, Displacements of the LBD tier induced by pH are summarized as mode vectors using Pymol (black line with red arrowhead). The consensus map at pH 8.0 and pH 5.5, which represent global average of all sub-classes, were used for comparison. The two structures were aligned at the M3 helices in the TMD which adopt virtually identical structures. LBD subunits are labeled as A-D. Pore central axis is indicated with a black diamond (top) and a broken arrow (bottom). **b**, The D1 lobes of the LBD tier, the subdomain closer to the NTD tier, at pH 8.0 and pH 5.5 are superimposed as described above. Red arrow: shortening of distance between the D1 lobes at pH 5.5. Inset: rearrangements of their G helices at Ala665.

# Reporting Summary

## Statistics

For all statistical analyses, confirm that the following items are present in the figure legend, table legend, main text, or Methods section.

| n/a | Confirmed | |
|---|---|---|
| ☐ | ☒ | The exact sample size (*n*) for each experimental group/condition, given as a discrete number and unit of measurement |
| ☐ | ☒ | A statement on whether measurements were taken from distinct samples or whether the same sample was measured repeatedly |
| ☐ | ☒ | The statistical test(s) used AND whether they are one- or two-sided *Only common tests should be described solely by name; describe more complex techniques in the Methods section.* |
| ☒ | ☐ | A description of all covariates tested |
| ☐ | ☒ | A description of any assumptions or corrections, such as tests of normality and adjustment for multiple comparisons |
| ☐ | ☒ | A full description of the statistical parameters including central tendency (e.g. means) or other basic estimates (e.g. regression coefficient) AND variation (e.g. standard deviation) or associated estimates of uncertainty (e.g. confidence intervals) |
| ☐ | ☒ | For null hypothesis testing, the test statistic (e.g. *F*, *t*, *r*) with confidence intervals, effect sizes, degrees of freedom and *P* value noted *Give P values as exact values whenever suitable.* |
| ☒ | ☐ | For Bayesian analysis, information on the choice of priors and Markov chain Monte Carlo settings |
| ☒ | ☐ | For hierarchical and complex designs, identification of the appropriate level for tests and full reporting of outcomes |
| ☒ | ☐ | Estimates of effect sizes (e.g. Cohen's *d*, Pearson's *r*), indicating how they were calculated |

*Our web collection on statistics for biologists contains articles on many of the points above.*

## Software and code

Policy information about availability of computer code

| Data collection | EEPU v2.6-3.3 (Thermofisher Scientific), pClamp10 (Molecular Devices) |
|---|---|
| Data analysis | Relion v4 and v5 (open source), CTFFIND4 (open source), motioncorr2 (open source), Resmap v1.95 (open source), coot v0.897 (for OSX, open source), chimera v1.14 (open source) , pymol v2.1 (Schrodinger), phenix v1.20 (open source), AlphaFold2, DynaMut2, Microsoft Excel v16 (Microsoft), Prism v10 (Graph Pad Software). |

For manuscripts utilizing custom algorithms or software that are central to the research but not yet described in published literature, software must be made available to editors and reviewers. We strongly encourage code deposition in a community repository (e.g. GitHub). See the Nature Portfolio guidelines for submitting code & software for further information.

## Data

Policy information about availability of data

All manuscripts must include a data availability statement. This statement should provide the following information, where applicable:
- Accession codes, unique identifiers, or web links for publicly available datasets
- A description of any restrictions on data availability
- For clinical datasets or third party data, please ensure that the statement adheres to our policy

Cryo-EM coordinates and corresponding EM-maps are deposited in the PDB and EMDB under the following accession codes; PDB:9B5Z (EMD-44232), PDB:9B60 (EMD-44233), PDB:9B67 (EMD-44248), PDB:9B68 (EMD-44249), PDB:9B6A (EMD-44251), PDB:9B69 (EMD-44250), PDB:9B61 (EMD-44234), PDB:9B63 (EMD-

44244), and PDB:9B64 (EMD-44245). Molecular dynamic (MD) simulation trajectories are deposited in Zenodo with the following DOI: 10.5281/zenodo.11654387. Request for materials (plasmids and cell lines) will be fulfilled for reasonable inquiries and should be addressed to Ingo Greger and Terunaga Nakagawa. Source data are provided with this paper.

## Human research participants

Policy information about studies involving human research participants and Sex and Gender in Research.

| | |
|---|---|
| Reporting on sex and gender | n/a |
| Population characteristics | n/a |
| Recruitment | n/a |
| Ethics oversight | n/a |

Note that full information on the approval of the study protocol must also be provided in the manuscript.

# Field-specific reporting

Please select the one below that is the best fit for your research. If you are not sure, read the appropriate sections before making your selection.

☒ Life sciences ☐ Behavioural & social sciences ☐ Ecological, evolutionary & environmental sciences

For a reference copy of the document with all sections, see nature.com/documents/nr-reporting-summary-flat.pdf

# Life sciences study design

All studies must disclose on these points even when the disclosure is negative.

| | |
|---|---|
| Sample size | Sample size for cryo-EM data collection was determined based on the knowledge that AMPARs require about 20-100 thousand particles to reach 3.5Å resolution. To to be able to sort out conformational variability of the NTDs by classifying itno 20-40 classes we estimated that about 1,000,000 ± 200,000 particles are needed, which translates into collecting about 20,000 micrographs (Zhang, Nature 2023, Nakagawa, NSMB 2024). <br> Electrophysiology sample sizes were determined based on literature review, previous experience with data of this sort, and reproducibility of results across independent experiments. The authors have extensive previous experience with data of this type (Zhang, Nature 2023&2021; Herguedas, Science 2019; Herguedas, Science 2016; Cais, Cell Reports 2014)., therefore sample sizes were based on understanding of sample variabilities. <br> Light microscopy sample sizes were determined based on previous FRAP experiments (Watson, eLife 2017) and were reproducible across recordings from multiple cells from three different culture preparations. |
| Data exclusions | CTF and rlnMaxResolution parameters were used to remove images with bad image quality. 3D classification removes particles based on objective statistical measures which retains particles with homogeneous structures containing high resolution signals. FRAP recordings were excluded or included based on cell viability and stability of conditions during recording. |
| Replication | Expression and purification were highly robust and reproducible across experiments. The half maps of the 3D refinement in each structure produced consistent results, which supports high consistency of data quality across micrographs. All electrophysiology data sets were pooled from at least two independent experiments and all results were successfully replicated at least five times. Light microscopy experiments were replicated from multiple cells across three different culture preparations. |
| Randomization | For Cryo-EM, division of datasets into two random halves was done based on standard approach in RELION. Randomization is not relevant to electrophysiology. Similarly for imaging experiments, the experimenter is in charge of handling plasmids, cell lines, transfection and acquiring the data on the microscope so it is not feasible to randomise. |
| Blinding | Blinding was not applicable to cryo-EM or MD simulations, because this type of study does not use group allocation. Researchers were not blinded for the acquisition or analysis of electrophysiology and imaging data as it was not technically or practically feasible to do so. Experimenter independence was ensured by application of defined exclusion criteria as stated above. |

# Reporting for specific materials, systems and methods

We require information from authors about some types of materials, experimental systems and methods used in many studies. Here, indicate whether each material, system or method listed is relevant to your study. If you are not sure if a list item applies to your research, read the appropriate section before selecting a response.

## Materials & experimental systems

| n/a | Involved in the study |
|-----|----------------------|
| ☐ | ☒ Antibodies |
| ☐ | ☒ Eukaryotic cell lines |
| ☒ | ☐ Palaeontology and archaeology |
| ☐ | ☒ Animals and other organisms |
| ☒ | ☐ Clinical data |
| ☒ | ☐ Dual use research of concern |

## Methods

| n/a | Involved in the study |
|-----|----------------------|
| ☒ | ☐ ChIP-seq |
| ☒ | ☐ Flow cytometry |
| ☒ | ☐ MRI-based neuroimaging |

## Antibodies

| | |
|---|---|
| Antibodies used | anti-FLAG M2 monoclonal antibody |
| Validation | Purchased from Sigma. The product is quality controlled. |

## Eukaryotic cell lines

Policy information about cell lines and Sex and Gender in Research

| | |
|---|---|
| Cell line source(s) | HEK293Tcells were purchased from ATCC and TetON HEK cell (Clontech) and their derivatives were isolated in Nakagawa lab. |
| Authentication | No further authentication of HEK293T was performed for cell lines used in the electrophysiology experiments. The TetON HEK cell line was purchased from Clontech. The cell morphology is spindle shaped and homogeneous. Growth rate was consistent with HEK cell. The cell line respond to DOX as described by the manufacturer. The cell line is sensitive to hygromycine and zeocin. The cell line is insensitive to G418. The line is used extensively in past literatures to generate stable cell lines that DOX dependently express proteins for structural studies. |
| Mycoplasma contamination | No mycoplasma testing was performed specifically for this study, the HEK293T cell line had been tested negative in the past. |
| Commonly misidentified lines (See ICLAC register) | HEK cells are listed in the register; however, our HEK cell lines come from reliable source (ATCC ) and are the only secondary cell type used in this study, which minimizes the risk of any cross-contamination. |

## Animals and other research organisms

Policy information about studies involving animals; ARRIVE guidelines recommended for reporting animal research, and Sex and Gender in Research

| | |
|---|---|
| Laboratory animals | C57/Bl6 mice of both sexes were used in this study at age postnatal day 0-1. Animals were housed with unlimited access to food and water under a standard 12 hour light-dark cycle, at normal room temperature (approx 20-22 degrees Centigrade). Pregnant mothers were monitored daily, and P0 refers to the day of litter discovery. |
| Wild animals | No wild animals were used in this study. |
| Reporting on sex | Dissociated cultures were prepared from pups of both sexes. There is no reported or discernible differences between sexes in electrophysiological properties of culture prepared at age P0-1. |
| Field-collected samples | No field collected samples were used in this study. |
| Ethics oversight | All procedures were carried out under PPL 70/8135 in accordance with UK Home Office regulations. Experiments conducted in the UK are licensed under the UK Animals (Scientific Procedures) Act of 1986 following local ethical approval. |

Note that full information on the approval of the study protocol must also be provided in the manuscript.

