## [Peer Review File · Nature Structural & Molecular Biology]

Peer Review Information

Manuscript Title: Proton-triggered rearrangement of the AMPA receptor N-terminal domains impacts receptor kinetics and synaptic localisation

Corresponding author name(s): Terunaga Nakagawa, Ingo Greger

Reviewer Comments & Decisions:

Decision Letter, initial version:
--

Message: 2nd May 2024

Dear Dr. Greger,

Thank you again for submitting your manuscript "Rearrangement of the GluA2 AMPA receptor N-terminal domain triggered by protons". We now have comments (below) from the 2 reviewers who evaluated your paper. In light of those reports, we remain interested in your study and would like to see your response to the comments of the referees, in the form of a revised manuscript.

You will see that while reviewers appreciate the results, they raise several concerns which will need to be addressed in a revision. Specifically, we agree with the reviewers that addressing the meaning of the results in the context of more physiological short-lasting fluctuations of pH would strengthen the manuscript. Please ensure to address the mismatch of the conditions between structural and functional experiments. Where further experimentation is not feasible, please make sure to address caveats in the text. Please be sure to address/respond to all concerns of the referees in full in a point-by-point response and highlight all changes in the revised manuscript text file. If you have comments that are intended for editors only, please include those in a separate cover letter.

We expect to see your revised manuscript within 12 weeks. If you cannot send it within this time, please contact us to discuss an extension; we would still consider your revision, provided that no similar work has been accepted for publication at NSMB or published elsewhere.

Reporting Summary:

When submitting the revised version of your manuscript, please pay close attention to our [Digital Image Integrity Guidelines](https://www.nature.com/nature-portfolio/editorial-policies/image-integrity). and to the following points below:

Please note that all key data shown in the main figures as cropped gels or blots should be presented in uncropped form, with molecular weight markers. These data can be aggregated into a single supplementary figure item. While these data can be displayed in a relatively informal style, they must refer back to the relevant figures. These data should be submitted with the final revision, as source data, prior to acceptance, but you may want to start putting it together at this point.

Data availability: this journal strongly supports public availability of data. All data used in accepted papers should be available via a public data repository, or alternatively, as Supplementary Information. If data can only be shared on request, please explain why in your Data Availability Statement, and also in the correspondence with your editor. Please note that for some data types, deposition in a public repository is mandatory - more information on our data deposition policies and available repositories can be found below: <https://www.nature.com/nature-research/editorial-policies/reporting-standards#availability-of-data>

Nature Structural & Molecular Biology is committed to improving transparency in authorship. As part of our efforts in this direction, we are now requesting that all authors identified as 'corresponding author' on published papers create and link their Open Researcher and Contributor Identifier (ORCID) with their account on the Manuscript Tracking System (MTS), prior to acceptance. This applies to primary research papers only. ORCID helps the scientific community achieve unambiguous attribution of all scholarly contributions. You can create and link your ORCID from the home page of the MTS by clicking on 'Modify my Springer Nature account'. For more information please visit please visit www.springernature.com/orcid.

[Redacted]

Sincerely,

Katarzyna Ciazynska, PhD
(she/her)
Associate Editor
Nature Structural & Molecular Biology
<https://orcid.org/0000-0002-9899-2428>

Referee expertise:

Referee #1: neurobiology

Referee #2: neurobiology, structural biology

Reviewers' Comments:

Reviewer #1:

Remarks to the Author:

The paper by Ivica et al. report about the protonation sensitivity of AMPARs with particular focus on the molecular rearrangements of the GluA2 subunit. The lab has a strong record on this NTD proportion of the AMPAR and applied consequently the available tools to define how changes in pH may impact on the activation, desensitisation and local membrane dynamics of AMPAR. The experiments are well done and support former physiological data that GluA2 containing AMPAR become less responsive in acidic environment and desensitize much faster. In addition the authors probed the idea whether AMPAR may also become more flexible in the acidic environment.

From a structural perspective this data are very solid and give further insights into the function of GluA2 containing AMPAR. There are many controls and conditions probed. Despite the mentioned publication in the end of the discussion I see here still a novelty in the reported data.

The physiological relevance is proposed by FRAP experiments in steady state conditions, means prolonged acidification of the extracellular solution, which most likely is only an exception within the brain during pathological conditions. Interesting would be whether short periods of local pH-drops are indeed effective to add to the dynamic behaviour of the GluA2 containing AMPAR? Bursting is the main code for hippocampal communication, thus transient changes in pH are probably much closer to physiology than prolonged changes in pH?

Reviewer #2:

Remarks to the Author:

Summary:

AMPA-type ionotropic glutamate receptors (AMPA receptors) mediate most of the excitatory transmission in the central nervous system. They are involved in various brain diseases and disorders and are promising drug targets. The GluA2 AMPA subtype is the major AMPAR expressed in the brain, and therefore, it is extremely important to understand their physiological and pathophysiological roles. Recently, Dr. Greger and his colleagues described the N-terminal domain (NTD) of AMPARs regulating the localization and gating of receptors subtype dependently (Zhang et al., 2023 Nature). In this original article from Ivica and colleagues, the authors further investigated the pH-dependent role of GluA2 NTD. They described the detailed conformational and functional modification and the receptor localization regulated by proton. Overall, this manuscript greatly advances our understanding of AMPAR gating and trafficking.

Significance:

- Previous studies have shown that the majority of native AMPA contains GluA2 subunit occupying the BD positions. Therefore, this discovery of the interaction at the GluA2 NTD dimer interface is fundamental to understanding the common regulation mechanism of native AMPA receptor complexes.
- Information on detailed pH-dependent NTD conformational changes and inter-protomer interaction is novel.
- The authors determined the relatively high-resolution structures of GluA2 with the compact or ruptured NTD at pH 8.0 and 5.5 in the absence of glutamate.
- The authors further studied the proton-regulated diffusion at the postsynaptic membrane.

Conclusions and suggested improvements:

Major comments:

1. Please describe whether protons (or NTD conformational changes) shift the EC50 value for glutamate activation of GluA2 AMPAR.
2. The way in which the conformational changes within the NTD are transduced to LBD and TMD is not very clear.
3. The analysis of receptor mobility at different pHs is interesting. Previous reports showed that the AMPAR NTD deletion reduces the surface expression of GluA1-4 AMPARs. Does the compact NTD formation also lead to receptor internalization (or reduced receptor surface expression)?
4. More details on the mechanism by which ruptured NTD speeds desensitization and slows recovery from a desensitized state from the structural perspective would be appreciated.

Minor comments :

1. The authors determined cryo-EM structures of GluA2 in complex with TARP $\gamma 2$ and further analyzed the local conformational changes by MD at pH 8.0 and pH 5.5. However, they conducted electrophysiological recordings and FRAP assay at pH 7.4 and 5.5. There is a mismatch between the conditions used for structural and functional analysis.
2. The structures of GluA2 at pH 5.5 showed heterogeneity of the LBD layer. Whether the protons directly binding to LBDs mainly induce LBD conformational changes and modulate the gating pH dependently, or the dynamics of NTD induce the conformational changes of LBD and alter the receptor gating, is not very clear.
3. Figure 1d: The authors show the sequence alignment of mouse AMPARs, but they use rat GluA2 AMPAR for cryo-EM studies (and perhaps for electrophysiological studies). It is more relevant to show the rat sequence alignment.
4. Please describe whether any conformational differences in the NTD (e.g. bi-lobe closure degree, dimer stability etc) were observed or not. Are NTD conformations identical in all classes?
5. Extended Data Fig. 3d: The title of the X-axis is not fully shown (I can read "Current (pA)", though).
6. Extended Data Fig. 3, p21 Line 522: Please add "e," between "(red)." and "Two".
7. Extended Data Fig. 7a: Showing particle numbers for each class would be helpful.
8. Extended Data Fig. 7b: The figures are tiny. Also, showing the distance between the COMs of two dimers might help us understand the differences between those classes.
9. P5, L119: Please add the abbreviate of MALS (Multi-Angle Light Scattering)
10. P6, L128: Please show the density of side residues (Fig. 1c, Extended Fig. 1c).
11. P8, L171: Perhaps Extended Data Fig 2d and Fig 1e?

12. P9, L193: Extended Data Figs. 3e ◊ Figure 3b?
13. P4 L77-78 & P14, L325-328: Please confirm that in the physiological environment, TARP $\gamma 2$ is not sufficient to anchor receptors at the active synaptic zone at acidic pH (and therefore, the NTD is the major factor which defines the receptor localization). Does proton also weaken the AMPA-TARP $\gamma 2$ interactions?
14. P15, L351-353: Please confirm that the authors suggested that the LBD D1-D1 rupture lead receptors in shallow desensitized states, and the additional NTD rupture at acidic pH further lead receptors in deeply desensitized states.
15. P15, L345: Extended Data Figs. 3e ◊ Probably Figure 3b?
16. P22 Figure 1a (and Figure 3d) It would be nice to add a description in the figure legends that they were recorded at pH 7.4 since this is the first figure (It is clear once I read the entire manuscript since it is color coded, though).

Data & methodology/ statistics/references
Appropriate.

Clarity and context:
Abstract, introduction and conclusion are all very clear.

Author Rebuttal to Initial comments

Ivica et al., Response to reviewers NSMB-A490

We thank the reviewers for the overall positive comments and helpful suggestions.

Reviewer #1:

Remarks to the Author:

The paper by Ivica et al. report about the protonation sensitivity of AMPARs with particular focus on the molecular rearrangements of the GluA2 subunit. The lab has a strong record on this NTD proportion of the AMPAR and applied consequently the available tools to define how changes in pH may impact on the activation, desensitisation and local membrane dynamics of AMPAR. The experiments are well done and support former physiological data that GluA2 containing AMPAR become less responsive in acidic environment and desensitize much faster. In addition the authors probed the idea whether AMPAR may also become more flexible in the acidic environment.

From a structural perspective this data are very solid and give further insights into the function of GluA2 containing AMPAR. There are many controls and conditions probed. Despite the mentioned publication in the end of the discussion I see here still a novelty in the reported data.

Thank you for these comments.

The physiological relevance is proposed by FRAP experiments in steady state conditions, means

prolonged acidification of the extracellular solution, which most likely is only an exception within the brain during pathological conditions. Interesting would be whether short periods of local pH-drops are indeed effective to add to the dynamic behaviour of the GluA2 containing AMPAR? Bursting is the main code for hippocampal communication, thus transient changes in pH are probably much closer to physiology than prolonged changes in pH?

This is an interesting point but is technically very challenging. We agree with the reviewer that such prolonged acidification may not occur during physiological synaptic transmission and the FRAP experiments are an extrapolation from physiological conclusions. However, cleft acidification is predicted to last only a fraction of a millisecond (Feghhi et al., Biophys J 2021), and is followed by prolonged periods of alkalinisation. We would be unable to detect these fluctuations due to technical limitations. With our microscope setup, imaging of one spine takes 1.26ms, exceeding the predicted time course of acidification (hundreds of microseconds; Feghhi 2021 study). As stability is required to maintain the spine in the field of view it was not possible to change solutions during the imaging time. Similarly, as SEP fluorescence changes at different pH it was not possible to change the pH after baseline fluorescence of the spine was recorded. Local release of low pH solution using a microelectrode pipette and single particle imaging may be able to capture GluA2 behaviour under more realistic pH changes in future experiments. For these reasons we opted for the *two independent steady-state* conditions to assess the impact of NTD splaying on AMPAR synaptic mobility by FRAP: low pH and the GluA2 F231A mutant (both exhibiting splayed NTDs), both of which exhibit similar diffusion rates, which are faster than GluA2 wt imaged at neutral pH. We have now increased our data set to further solidify these data, and have also included data for dendritic spines, as outside the synapse no anchoring would be expected (and this is what we show). These additional data are included in Fig. 6a of the revision. We also discuss the bidirectional pH switch accompanying vesicular release, which has recently been assessed in a careful modelling study (Feghhi et al., Biophys J 2021), which we cite in the revised discussion section.

Reviewer #2:

Remarks to the Author:

Summary:

AMPA-type ionotropic glutamate receptors (AMPA) mediate most of the excitatory transmission in the central nervous system. They are involved in various brain diseases and disorders and are promising drug targets. The GluA2 AMPA subtype is the major AMPAR expressed in the brain, and therefore, it is extremely important to understand their physiological and pathophysiological roles. Recently, Dr. Greger and his colleagues described the N-terminal domain (NTD) of AMPARs regulating the localization and gating of receptors subtype

independently (Zhang et al., 2023 Nature). In this original article from Ivica and colleagues, the authors further investigated the pH-dependent role of GluA2 NTD. They described the detailed conformational and functional modification and the receptor localization regulated by proton. Overall, this manuscript greatly advances our understanding of AMPAR gating and trafficking.

Thank you for these positive comments.

Significance:

- Previous studies have shown that the majority of native AMPA contains GluA2 subunit occupying the BD positions. Therefore, this discovery of the interaction at the GluA2 NTD dimer interface is fundamental to understanding the common regulation mechanism of native AMPA receptor complexes.
- Information on detailed pH-dependent NTD conformational changes and inter-protomer interaction is novel.
- The authors determined the relatively high-resolution structures of GluA2 with the compact or ruptured NTD at pH 8.0 and 5.5 in the absence of glutamate.
- The authors further studied the proton-regulated diffusion at the postsynaptic membrane.

Conclusions and suggested improvements:

Major comments:

1. Please describe whether protons (or NTD conformational changes) shift the EC50 value for glutamate activation of GluA2 AMPAR.

Thank you for this suggestion. We have now performed these additional experiments. We do observe a rightward shift in the L-glutamate dose-response curve in acidic versus alkaline pH, suggesting lower agonist efficacy at low pH (Fig .1; below), which likely stems from the LBD rearrangements we observe structurally. This further demonstrates the wide range of consequences of pH for AMPAR function. These data are now included on p 8 and are incorporated into Extended Data Fig. 3d of the revised paper.

Fig. 1 L-glutamate dose-response relationship of GluA2/TARPy2 at neutral pH (7.4; black data points) and under acidic conditions (pH 5.5; red data points). A significant rightward shift, resulting in a 7-fold increased EC_{50} is observed at acidic pH in whole cell recordings.

2. The way in which the conformational changes within the NTD are transduced to LBD and TMD is not very clear.

Thank you for pointing this out. Together with our recent study (Zhang et al., Nature 2023), our data suggest that rupture of the NTD layer (through H208 protonation at acidic pH) destabilises the entire receptor assembly, i.e. has a knock-on effect on the intrinsically unstable LBD layer, which will contribute to the functional changes we describe. In Zhang et al (2023), we had shown that rupture of the tetrameric NTD, triggered by the GluA2 F231A mutation in the NTD BD interface, is coupled to greater conformational flexibility in the LBD tier, and this is also evident in GluA1 (with its inherently unstable NTD tier (Zhang et al., Nature 2023)). Our current study further supports these observations and suggest that a natural trigger, protons, can cause this in GluA2-containing receptors.

We have now added a sentence in the discussion on p. 15 to clarify this.

3. The analysis of receptor mobility at different pHs is interesting. Previous reports showed that the AMPAR NTD deletion reduces the surface expression of GluA1-4 AMPARs. Does the compact NTD formation also lead to receptor internalization (or reduced receptor surface expression)?

Albeit an interesting possibility, addressing this point would require a whole new set of experiments, which could be envisaged for future studies. Based on our current data we would

expect that a compact NTD would decrease endocytosis, resulting from reduced anchoring at synaptic sites, enabling diffusion to endocytic zones at the edge of the PSD.

4. More details on the mechanism by which ruptured NTD speeds desensitization and slows recovery from a desensitized state from the structural perspective would be appreciated.

This further relates to the 2nd point above, which is now discussed on p 15 and in the Results section on p 13. Briefly, based on structure/function relationships of the GluA2 F231A mutant (which exhibits a splayed NTD layer akin to GluA2 in acidic pH) and on GluA1 (which exhibits a highly mobile NTD layer; Zhang et al., Nature 2023), instability of the NTD tier results in greater flexibility of the LBDs. Relieved from constraint by the BD NTD interface, the LBDs can sample a wider conformational space. Under desensitizing conditions, NTD splaying enables rupture of the otherwise intact LBD dimers, leading to deep desensitization and thus slowed recover from these states (which require re-dimerization of the LBDs for receptor activation).

Minor comments :

1. The authors determined cryo-EM structures of GluA2 in complex with TARP γ 2 and further analyzed the local conformational changes by MD at pH 8.0 and pH5.5. However, they conducted electrophysiological recordings and FRAP assay at pH 7.4 and 5.5. There is a mismatch between the conditions used for structural and functional analysis.

For the functional experiments in cells (electrophysiology and FRAP) we chose a neutral pH, close to physiological conditions (pH 7.4). For the structural studies we wanted to ensure a clear separation between acidic and alkaline conditions, and therefore raised the pH to 8.0. To address this point experimentally, we now conducted additional recordings of GluA2/TARP γ 2 at pH 7.4 versus 8.0, alongside pH 5.5 (Fig. 2; below), using fast L-glutamate application on isolated HEK293 membrane patches. We *do not* observe any difference between pH 7.4 and 8.0. These new data are now described on page 12 and are shown in Extended Data Fig. 6a.

Fig. 2: Comparison of the peak response (left panel), entry into desensitization kinetics (middle panel), and recovery kinetics from desensitization (right panel). Three different pH conditions were probed as indicated (pH 8 = purple data points, pH 7.4 grey data points and pH 5.5 in red). No difference in these parameters were apparent between pH 8.0 and 7.4.

2. The structures of GluA2 at pH 5.5 showed heterogeneity of the LBD layer. Whether the protons directly binding to LBDs mainly induce LBD conformational changes and modulate the gating pH dependently, or the dynamics of NTD induce the conformational changes of LBD and alter the receptor gating, is not very clear.

This is difficult to distinguish. We would argue that in addition to causing NTD splaying (via protonation of H208), protons also directly target the LBD, which is suggested by the pH sensitivity of the GluA2 mutant lacking the NTD (Fig. 3 b and c). Hence, in addition to enabling greater LBD flexibility caused by NTD splaying, these data suggest that protons directly target the LBD layer.

3. Figure 1d: The authors show the sequence alignment of mouse AMPARs, but they use rat GluA2 AMPAR for cryo-EM studies (and perhaps for electrophysiological studies). It is more relevant to show the rat sequence alignment.

This region is highly conserved across mammalian AMPAR subunits. Hence, rat and mouse (and human) sequences are identical. Below is a section of the Promals alignment around GluA2 H208 and F231, documenting this.

Fig. 4: Cryo-EM particle number within each 3D class at pH 5.5 (indicated above the maps).

8. Extended Data Fig. 7b: The figures are tiny. Also, showing the distance between the COMs of two dimers might help us understand the differences between those classes.

We now enlarged the panel in Extended Data Fig. 7, and provide the COM between the BD NTDs of the pH 5.5 classes (the BD distance in the pH 8.0 classes is consistently ~ 55 Å; Fig. 6 below):

Fig. 5: Models of GluA2/TARPy2 from the pH 5.5 3D classes. Top row side view onto NTD and LBD tiers. The class number and center of mass (COM) between the BD chains is indicated. Bottom row: Top view onto the NTD tier.

9. P5, L119: Please add the abbreviate of MALS (Multi-Angle Light Scattering).

Done, thank you.

10. P6, L128: Please show the density of side residues (Fig. 1c, Extended Fig. 1c).

The model in this figure came from a high-resolution crystal structure (PDB 3H5V), which we now clarify in the figure legend. This remains the highest resolution view of this interface and is therefore best suited to document interface arrangement.

In the revised paper, we now show the BD NTD interface from our structure determined at pH 8.0 (Extended data Fig. 7a):

11. P8, L171: Perhaps Extended Data Fig 2d and Fig 1e?

Thank you for spotting this, now corrected.

12. P9, L193: Extended Data Figs. 3e ◊ Figure 3b?

Thank you for spotting this, now corrected.

13. P4 L77-78 & P14, L325-328: Please confirm that in the physiological environment, TARP $\gamma 2$ is not sufficient to anchor receptors at the active synaptic zone at acidic pH (and therefore, the NTD is the major factor which defines the receptor localization). Does proton also weaken the AMPA-TARP $\gamma 2$ interactions?

That is an interesting point. According to our current understanding, both the TARP and the NTD contribute to synaptic anchoring (Watson et al., Nat Comms 2021; reviewed in Stockwell et al., Bioessays 2024). We hypothesise that while the TARP, via cytosolic interactions, locates the

receptor to the PSD per se (work from Roger Nicoll, David Brecht and others), the NTD enables *subs synaptic location* possibly beneath presynaptic release sites. Accordingly, weakening the NTD anchor (resulting from splaying) by protonation of the cleft would destabilise receptor positioning from release sites, but will not affect cytosolic interaction with the PSD. Furthermore, based on our structural studies the TARP does not appear to dissociate from the receptor, both at low pH and under desensitizing conditions (Herguedas et al., Nat Comms 2022; Zhang et al., Nature 2023).

14. P15, L351-353: Please confirm that the authors suggested that the LBD D1-D1 rupture lead receptors in shallow desensitized states, and the additional NTD rupture at acidic pH further lead receptors in deeply desensitized states.

Yes, we suggest that reorganisation of the LBD D1 interface in LBD dimers leads to shallow desensitization, while rupture of the LBDs into monomers, which is facilitated by NTD splaying (Zhang et al., Nature 2023 and outlined above) leads to deep desensitized states. We have added a sentence at this position in the discussion to make this clearer: 'In both cases, GluA2_{F231A} and GluA1, desensitization is accompanied by rupture of the LBD dimers into monomers, which is not seen in desensitized GluA2, where the LBDs rearrange but remain dimeric^{38,55}.'

15. P15, L345: Extended Data Figs. 3e ◊ Probably Figure 3b?

Yes, it should cite Fig 3b, thank you.

16. P22 Figure 1a (and Figure 3d) It would be nice to add a description in the figure legends that they were recorded at pH 7.4 since this is the first figure (It is clear once I read the entire manuscript since it is color coded, though).

Done, thank you.

Decision Letter, first revision:

Message: Our ref: NSMB-A49032A

30th May 2024

Dear Dr. Greger,

Thank you for submitting your revised manuscript "Proton-triggered rearrangement of the AMPA receptor N-terminal domains impacts receptor kinetics and synaptic localisation" (NSMB-A49032A). It has now been seen by the original referees and their comments are below. The reviewers find that the paper has improved in revision, and therefore we'll be happy in principle to publish it in Nature Structural & Molecular Biology, pending minor revisions to satisfy the referees' final requests and to comply with our editorial and formatting guidelines.

We are now performing detailed checks on your paper and will send you a checklist detailing our editorial and formatting requirements in about 2-3 weeks. Please do not upload the final materials and make any revisions until you receive this additional information from us.

Sincerely,

Katarzyna Ciazynska, PhD
(she/her)
Associate Editor
Nature Structural & Molecular Biology
<https://orcid.org/0000-0002-9899-2428>

Reviewer #1 (Remarks to the Author):

Within the revised version the authors could sustain their finding that transient changes in the extracellular pH do impact on the ECD of AMPAR and have potential impact in their synaptic organisation.

The additional data provided by the authors in Fig.6 support a synapse specific impact of the pH-sensitivity, described in the paper. The physiological relevance is still a hypothetical suggestion, but traceable arguments are given to not include more experiments into the paper. I do have no further comments to the manuscript.

Reviewer #2 (Remarks to the Author):

The authors have addressed all of my concerns about the original manuscript. The revised manuscript is ready for publication.

Final Decision Letter:

Message: 8th Jul 2024

Dear Dr. Greger,

We are now happy to accept your revised paper "Proton-triggered rearrangement of the

AMPA receptor N-terminal domains impacts receptor kinetics and synaptic localisation" for publication as an Article in Nature Structural & Molecular Biology.

Your paper will be published online soon after we receive proof corrections and will appear in print in the next available issue. You can find out your date of online publication by contacting the production team shortly after sending your proof corrections.

You may wish to make your media relations office aware of your accepted publication, in case they consider it appropriate to organize some internal or external publicity. Once your paper has been scheduled you will receive an email confirming the publication details. This is normally 3-4 working days in advance of publication. If you need additional

notice of the date and time of publication, please let the production team know when you receive the proof of your article to ensure there is sufficient time to coordinate. Further information on our embargo policies can be found here: <https://www.nature.com/authors/policies/embargo.html>

Please note that *Nature Structural & Molecular Biology* is a Transformative Journal (TJ). Authors may publish their research with us through the traditional subscription access route or make their paper immediately open access through payment of an article-processing charge (APC). Authors will not be required to make a final decision about access to their article until it has been accepted. Find out more about Transformative Journals

Sincerely,

Katarzyna Ciazynska, PhD
(she/her)
Associate Editor
Nature Structural & Molecular Biology
<https://orcid.org/0000-0002-9899-2428>